# Evolution of lateralized gustation in nematodes

Marisa Mackie[1], Vivian Vy Le[1], Heather R Carstensen[1], Nicole R Kushnir[1], Dylan L Castro[1], Ivan M Dimov[1], Kathleen T Quach[2], Steven J Cook[3†], Oliver Hobert[3], Sreekanth H Chalasani[2], Ray L Hong[1*]

[1]Department of Biology, California State University, Northridge, Northridge, United States; [2]Molecular Neurobiology Laboratory, Salk Institute for Biological Studies, La Jolla, United States; [3]Department of Biological Sciences, Howard Hughes Medical Institute, Columbia University, New York, United States

**\*For correspondence:**
ray.hong@csun.edu

**Present address:** [†]Neural Coding Department, Allen Institute for Brain Science, Seattle, United States

**Competing interest:** The authors declare that no competing interests exist.

## eLife Assessment

Mackie and colleagues present a **valuable** comparison of lateralized gustation in two well-studied nematodes. Their results present **convincing** evidence that ASEL/R lateralization exists and is achieved by different means in P. pacificus compared to *C. elegans*. This work will be of interest to neurobiologists interested in how small nervous systems make sense of the environment, and how evolution can take multiple paths to asymmetry within a neuron class.

**Abstract** Animals with small nervous systems have a limited number of sensory neurons that must encode information from a changing environment. This problem is particularly exacerbated in nematodes that populate a wide variety of distinct ecological niches but only have a few sensory neurons available to encode multiple modalities. How does sensory diversity prevail within this constraint in neuron number? To identify the genetic basis for patterning different nervous systems, we demonstrate that sensory neurons in *Pristionchus pacificus* respond to various salt sensory cues in a manner that is partially distinct from that of the distantly related nematode *Caenorhabditis elegans*. Previously we showed that *P. pacificus* likely lacked bilateral asymmetry (Hong et al., 2019). Here, we show that by visualizing neuronal activity patterns, contrary to previous expectations based on its genome sequence, the salt responses of *P. pacificus* are encoded in a left/right asymmetric manner in the bilateral ASE neuron pair. Our study illustrates patterns of evolutionary stability and change in the gustatory system of nematodes.

## Introduction

Nematodes form a vast array of ecological relationships, from specialized parasite–host dependencies to nematode–microbial interactions, each one demanding exquisitely fine-tuned sets of sensory palates that span multiple modalities (*Bargmann, 2006*; *Hong and Sommer, 2006*; *Chaisson and Hallem, 2012*; *Wheeler et al., 2020*; *Rengarajan and Hallem, 2016*; *Koneru et al., 2016*; *Lo and Sommer, 2022*; *Lo et al., 2024*). Yet, the number of sensory neurons across diverse nematode species seems to be constrained (*White et al., 1986*; *Schafer, 2016*; *Hong et al., 2019*). Several free-living and parasitic nematode species examined by serial section electron microscopy have nearly identical numbers of 12–13 pairs of head sensory neurons, known as the amphid neurons (*Hong et al., 2019*; *Ward et al., 1975*; *Li et al., 2001*; *Bumbarger et al., 2009*; *Zhu et al., 2011*). How does sensory diversity arise within this constraint in neuron number? When coupled with well-described neuronal anatomy, this conserved neuron count allows for detailed comparisons at the single-cell resolution

and represents an opportunity to interrogate how sensory cues are processed by anatomically similar nervous systems to produce species-specific or developmental stage-dependent behavioral outputs. To identify the genetic basis for patterning different nervous systems and to understand the processes that underlie evolutionary changes in adapting to different environments, several comparative model systems have been developed to promote comparisons to the well-studied nematode *Caenorhabditis elegans* at the genetic and cellular levels, including the predatory entomophilic nematode, *Pristionchus pacificus* (*Hong et al., 2019*; *Loer and Rivard, 2007*; *Baiocchi et al., 2017*; *Gang et al., 2020*; *Bryant et al., 2022*).

As expected from their association with insects in the wild, the olfactory preferences of *P. pacificus* are distinct from those of *C. elegans* and the human parasite *Strongyloides stercoralis* (*Hong and Sommer, 2006*; *Chaisson and Hallem, 2012*; *Hallem et al., 2011*), but little is known about *P. pacificus* responses to water-soluble compounds. In *C. elegans*, the main salt receptor neuron class comprises of a bilateral pair of left and right ASE neurons (ASEL and ASER), which serve to induce an attractive locomotory response toward an increase in salt concentration (*Bargmann and Horvitz, 1991*). The gene *che-1* (*che*mosensory defective) encodes a transcription factor that is exclusively expressed in the ASE neurons and is required for their proper differentiation, such that a *che-1* mutant results in defective salt attraction (*Dusenbery et al., 1975*; *Uchida et al., 2003*; *Chang et al., 2003*; *Etchberger et al., 2007*).

One major role of *C. elegans* CHE-1 is to promote lateral asymmetry in the ASE neurons. The left and right ASE neurons asymmetrically express receptor-type guanylyl cyclases (rGCs, encoded by *gcy* genes) (*Yu et al., 1997*). This finding led to the realization that the ASE neurons are lateralized, such that the left and the right ASE neurons differentially respond to distinct salt ions (*Pierce-Shimomura et al., 2001*; *Chang et al., 2004*; *Suzuki et al., 2008*; *Ortiz et al., 2009*). This observation led in turn to the identification of a complex gene regulatory network that genetically programs the distinct sensory potentials of the left and right ASE neurons (*Hobert, 2014*), which includes a miRNA, *lsy-6*, at the top of this gene regulatory network (*Hobert, 2014*; *Johnston and Hobert, 2003*; *Cochella and Hobert, 2012*). However, the *lsy-6* miRNA evolved selectively in the *Caenorhabditis* genus (*Ahmed et al., 2013*) and is absent in *P. pacificus*. Moreover, *P. pacificus* does not show an expansion of the ASEL-type and ASER-type rGCs, as *C. elegans* does (*Hong et al., 2019*). With these two genomic observations in mind, we had previously proposed that the ASE neurons are unlikely to be lateralized in *P. pacificus* (*Hong et al., 2019*).

However, we now revise this view in light of our work on mapping chemosensory responses in *P. pacificus* on the level of behavior and neuronal activity. We demonstrate that *P. pacificus* does in fact show lateralized chemosensory profiles, indicating that *P. pacificus* must have evolved independent means to establish ASE laterality. We also show that the tastant palate of *P. pacificus* is distinct from that of *C. elegans*, and that its dependence on ASE, as well as its terminal selector transcription factor *che-1*, have also diverged.

## Results

### *P. pacificus* and *C. elegans* differ in their behavioral responses to salts

Previous cross-species comparisons between *P. pacificus* and *C. elegans* indicated strong differences in olfaction preferences that reflect their divergent evolutionary histories and ecology (*Hong and Sommer, 2006*). To identify the neurons that mediate gustation, we first compared the chemosensory profiles of these two species toward water-soluble ions. In this survey, we found ammonium salts to be the strongest attractants to wildtype *P. pacificus* J4 to adult hermaphrodites ($NH_4Br$, $NH_4Cl$, and $NH_4I$), with $NH_4I$ significantly more attractive to *P. pacificus* compared to *C. elegans* (*Figure 1*). Notably, in contrast to *C. elegans* (*Ward, 1973*), we find that *P. pacificus* is less attracted to NaCl and LiCl compared to the ammonium salts (*Figure 1*). Also, *P. pacificus* is repulsed by acetate salts (NaAc and $NH_4Ac$), which induce attractive responses in *C. elegans* (*Frøkjaer-Jensen et al., 2008*). We conclude that *P. pacificus* and *C. elegans* display differences in their salt preferences.

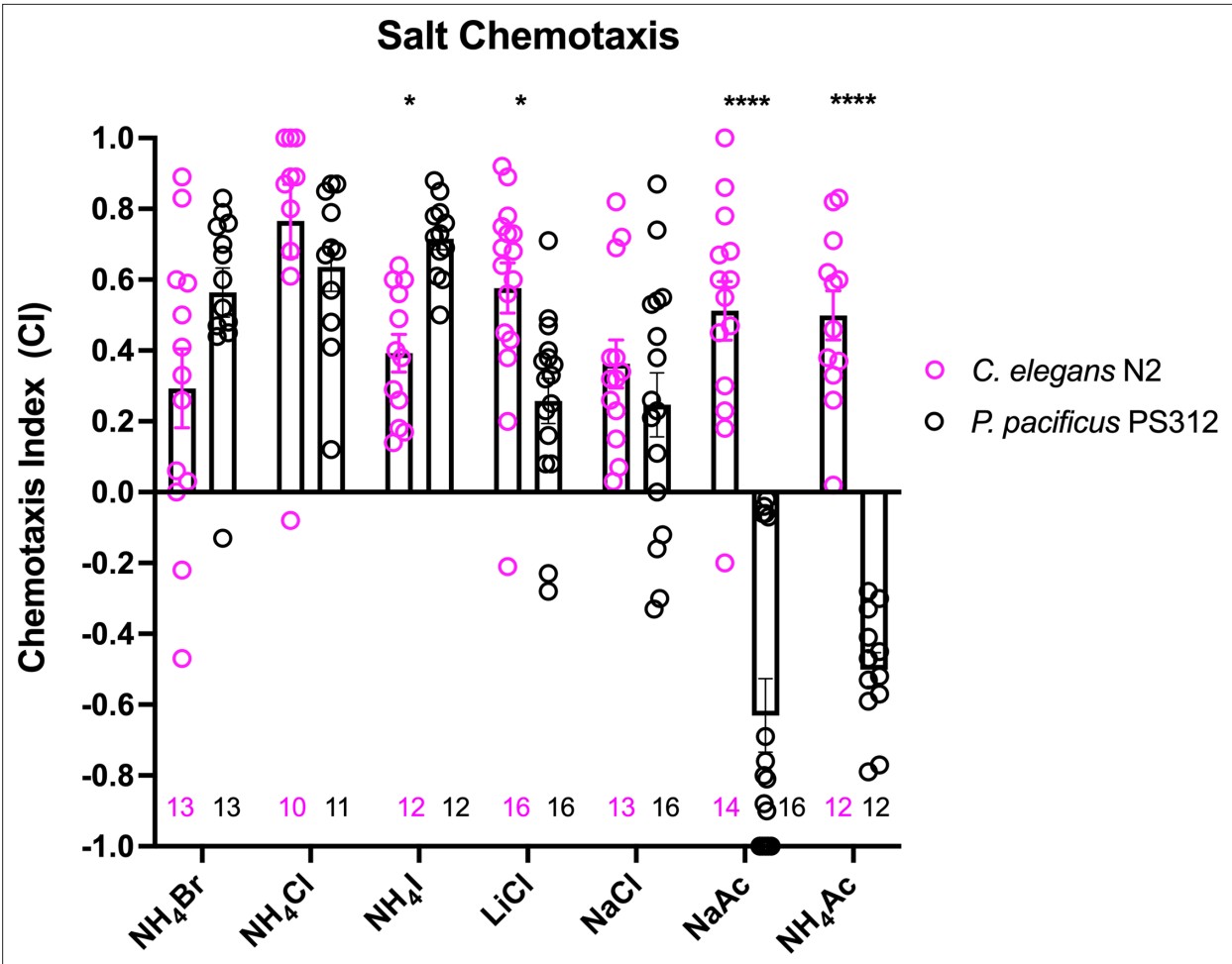

**Figure 1.** A comparison of chemotaxis responses to water-soluble ions between *P. pacificus* and *C. elegans*. J4 to adult hermaphrodites from the two species responded to NH₄I, LiCl, and acetates significantly differently. Using two-way ANOVA, significant difference found between wildtype *P. pacificus* and *C. elegans* for the same salt is indicated above each pair (*p < 0.05, ****p < 0.0001), while the differences within *P. pacificus* is as follows: all salts showed difference when compared to NaAc and to NH₄Ac (****p < 0.0001), but not between NH₄Ac and NaAc. Both LiCl and NaCl attraction are significantly lower than NH₄Cl (*p < 0.05) and NH₄I (***p < 0.001). Error bars denote standard error of the mean and the sample sizes are indicated on the bottom.

### *Ppa-che-1* shows similarities and differences to *Cel-che-1* in both expression and function

The *C. elegans che-1* mutant (<u>che</u>motaxis-defective) was originally isolated for its inability to respond to a broad panel of salt tastants (*Dusenbery et al., 1975*; *Ward, 1973*), including those described above (*Suzuki et al., 2008*; *Ortiz et al., 2009*; *Frøkjaer-Jensen et al., 2008*). *che-1* was found to encode for a Zn finger transcription factor that is exclusively expressed in the ASE neuron pair (*Etchberger et al., 2007*), which through laser ablations had been found to be the main salt receptor neurons (*Bargmann et al., 1993*). *che-1* was found to control the entire differentiation of the ASE neurons, including the expression of putative receptors of the GCY receptor guanylyl cyclase family (*Uchida et al., 2003*; *Chang et al., 2003*; *Etchberger et al., 2007*; *Etchberger et al., 2009*).

To assess whether *che-1* performs a similar function in salt perception for *P. elegans*, we analyzed the expression and function of the *Ppa-che-1* ortholog. In a previous paper, we reported that the 5' region of the sole *Pristionchus che-1* ortholog directs reporter expression to the *Ppa* ASE and *Ppa* ASG neuron classes (*Hong et al., 2019*). Re-examination of our provisional cell identifications using a newly generated *che-1p::GFP* strain with stronger neurite expression revealed highly elaborated finger-like dendritic endings in the more anterior amphid neuron that could

unambiguously be assigned to the AFD neurons (*Figure 2A–C*), prompting us to reassign expression of *che-1* to ASE and AFD.

We confirmed that the *che-1* reporter transgene indicates the full expression of the endogenous *che-1* locus by tagging the endogenous *che-1* locus with an ALFA-tag (*Igreja et al., 2022*). The *che-1::ALFA* animals showed staining in 2 pairs of head neurons whose position is consistent with being the ASE and AFD neurons (*Figure 2D, E*). By crossing the *che-1* reporter transgene into a *che-1* mutant background (see below), we also found that *che-1* autoregulates its own expression especially in the ASE neurons (*Figure 2F*, GFP expression in 97% in WT (*n* = 36) vs 4% in *che-1(−)* (*n* = 48)), as it does in *C. elegans* (*Etchberger et al., 2007*). The reduction of *che-1p::GFP* in the AFDs was also observed (GFP expression in 100% in WT (*n* = 36) vs 33% in *che-1(−)* (*n* = 48)).

To provide further evidence that *Ppa*-CHE-1 is indeed expressed in the *Ppa*-AFD neurons, we analyzed the expression of the *Ppa-ttx-1p::RFP* reporter. In *C. elegans*, the OTX homeodomain transcription factor TTX-1 is a terminal selector expressed in the AFD neurons required for designating the AFD fate (*Satterlee et al., 2001*). We found *ttx-1p::RFP* to be strongly expressed in a pair of neurons with the hallmark finger-like dendritic ending of the AFD neurons (*Figure 2G, H*), as well as expression in other head and tail neurons (possibly RIP and RIB) and cells likely to be the pharyngeal marginal cells based on likely conservation with *Cel-ttx-1* expression (*Reilly et al., 2022*). To confirm that endogenous TTX-1 protein expression is co-expressed with *che-1p::GFP*, we examined C-terminally tagged *ttx-1:ALFA* animals and found co-localization in the same anterior pair of amphid neurons, but no co-expression in the posterior pair of amphid neurons (*Figure 2I, J*, *n* = 15). Interestingly, the promoter expression in the posterior pair of amphid neurons in animals with both *ttx-1p::RFP* and *gcy-22.3p::GFP* reporters do co-localize in the ASER neurons (*Figure 2—figure supplement 1*), possibly due to differences between the expression patterns of the four possible *ttx-1* splice forms. The transcriptional and protein co-expression of *ttx-1* and *che-1* in the AFD neurons unequivocally show the expression of *che-1* in both AFD and ASE neurons in *P. pacificus*.

Next, we examined whether *P. pacificus* salt responsiveness shows similar *che-1* dependence as in *C. elegans*. We generated two putative null alleles in the *Ppa-che-1* homolog using CRISPR/Cas9 genome engineering, through introduction of small deletions in the first exon of the gene, thereby resulting in frameshift and premature stops (*Figure 3A*; *Figure 3—figure supplement 2*). Both *Ppa-che-1* alleles exhibited defects in attraction toward $NH_4Cl$ and LiCl compared to wildtype (*Figure 3B*). However, unlike in *C. elegans*, *Ppa-che-1* mutants showed no detectable difference in responses to $NH_4I$, NaCl, and NaAc.

We considered two different possibilities for the behavioral differences of *Ppa-che-1* and *Cel-che-1* mutants. *P. pacificus* may use sensory neurons other than ASE to sense these cues, or alternatively, *Ppa-che-1* may not have the same fundamental impact on ASE function in *P. pacificus* as it does in *C. elegans*. To explore these different possibilities, we silenced *che-1* expressing neurons by expressing codon-optimized HisCl1 channel under control of the *che-1* promoter. Histamine-treated *che-1p::HisCl1* animals showed complete loss of attraction to $NH_4Br$, $NH_4Cl$, and $NH_4I$ but did not significantly alter their repulsive response to NaAc (*Figure 3C*). As a control, we show that the presence of the *che-1p::HisCl1* transgene was necessary for the knockdown of $NH_4Br$ attraction (*Figure 3—figure supplement 1*). These findings corroborate that NaAc is sensed by neurons other than ASE (or AFD, in which the *che-1* promoter also drives HisCl1). Since $NH_4I$ sensation is affected by silencing of *che-1(+)* neurons but is unaffected in *che-1* mutants, ASE function may be more greatly impacted by the silencing of ASE than by the loss of *che-1*.

## The ASE neurons show left/right asymmetric responses to salt

To assess whether *Ppa* ASE neurons show the same lateralized response to salt ions as *Cel* ASE neurons, we generated transgenic *P. pacificus* lines that express RCaMP in ASE neurons and assessed calcium responses to attractive salts (*Figure 4*, *Figure 4—figure supplements 1 and 2*). Specifically, we looked for changes in calcium levels immediately after the addition and removal of specific salts. When 250 mM $NH_4Cl$ is administered, an 'ON' response is observed as calcium transiently increases in the left ASE neuron (ASEL). In contrast, an 'OFF' response was observed as calcium sharply dips in the right ASE neuron (ASER) before quickly returning to baseline when the salt was removed, which suggests hyperpolarization of this neuron (*Figure 4A, B*; *Suzuki et al., 2008*). However, when presented with a tenfold lower concentration of 25 mM $NH_4Cl$, the 'OFF' response completely

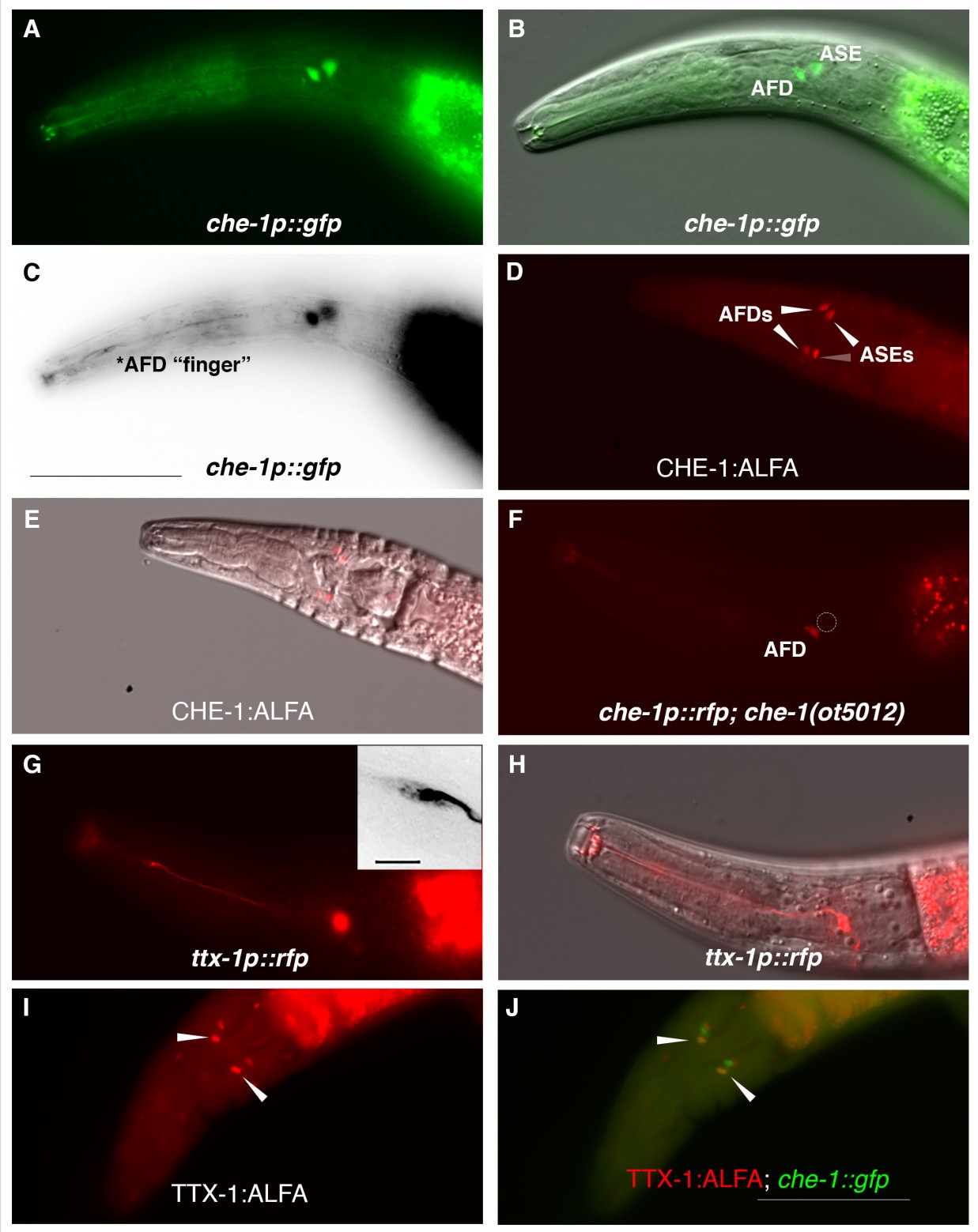

**Figure 2.** *P. pacificus che-1* expression in ASE and AFD amphid neurons. (**A, B**) The *che-1::GFP* marker in the *che-1::RCaMP* reporter strain is expressed in the ASE and AFD amphid neurons. (**C**) *che-1::GFP* expression is detectable in the morphologically distinct AFD neurons with 'finger'-like dendritic endings. (**D, E**) Immunostaining of *CHE-1::ALFA* shows two pairs of amphid neuron cell bodies corresponding to the ASE and AFD neurons; dorsal–ventral view (n = 114). (**F**) The loss of *che-1* results in loss *che-1::RFP* expression in the ASE (circle) while retaining reduced AFD expression. (**G**) *ttx-1p::RFP* expression in the AFD neurons with 'finger'-like dendritic endings. Inset shows expanded inverted black–white image of the AFD dendritic

*Figure 2 continued on next page*

*Figure 2 continued*

ending. (**H**) AFD expression of the same *ttx-1::RFP* animal shown in (**G**) in a different plane with cell body in focus. (**I, J**) Immunostaining of *TTX-1::ALFA* (red) shows one pair of amphid neuron cell bodies co-localizing with the anterior pair of *che-1::GFP*-expressing AFD neurons (yellow); dorsal–ventral view (*n* = 13). Anterior is left and the scale bar in (**C**) represents 50 µm for all panels except for the G inset, which is 5 µm.

The online version of this article includes the following figure supplement(s) for figure 2:

**Figure supplement 1.** Co-expression of *gcy-22.3p::GFP* and *ttx-1p::RFP* in ASER.

disappeared in ASER while the 'ON' response became more pronounced in ASEL. The ASER responses to 250 and 25 mM NaCl (*Figure 4D*) were very similar to the 'OFF' response (including hyperpolarization) observed for NH$_4$Cl, but the ASEL responses differ between the two salts: instead of the 'ON' response expected of ASEL, we observed a relatively weak 'OFF' response without the characteristic hyperpolarization but accompanied by an attenuated 'bump', a profile that we classify as an 'OFF-2' response (*Figure 4C*). Finally, we examined the response to NH$_4$I and found that it also elicited laterally asymmetric responses, but yet again distinct from the responses to both NH$_4$Cl and NaCl, with ASEL showing a strong 'ON' response, and ASER showing an 'ON–OFF' biphasic response to 250 mM NH$_4$I (*Figure 4E, F*; *Kato et al., 2014*; *Wang et al., 2015*). Interestingly, the ASER exhibited an 'OFF'-only response to 25 mM NH$_4$I, which was not observed for the same concentration of NH$_4$Cl and NaCl, and thus may reflect the higher response to NH$_4$I than to NH$_4$Cl and NaCl in the behavior assays. Altogether, *P. pacificus* ASE neurons clearly show left/right asymmetric responses to salt attractants and these asymmetric responses show similarities and differences to the *C. elegans* ASE taste neurons (see Discussion).

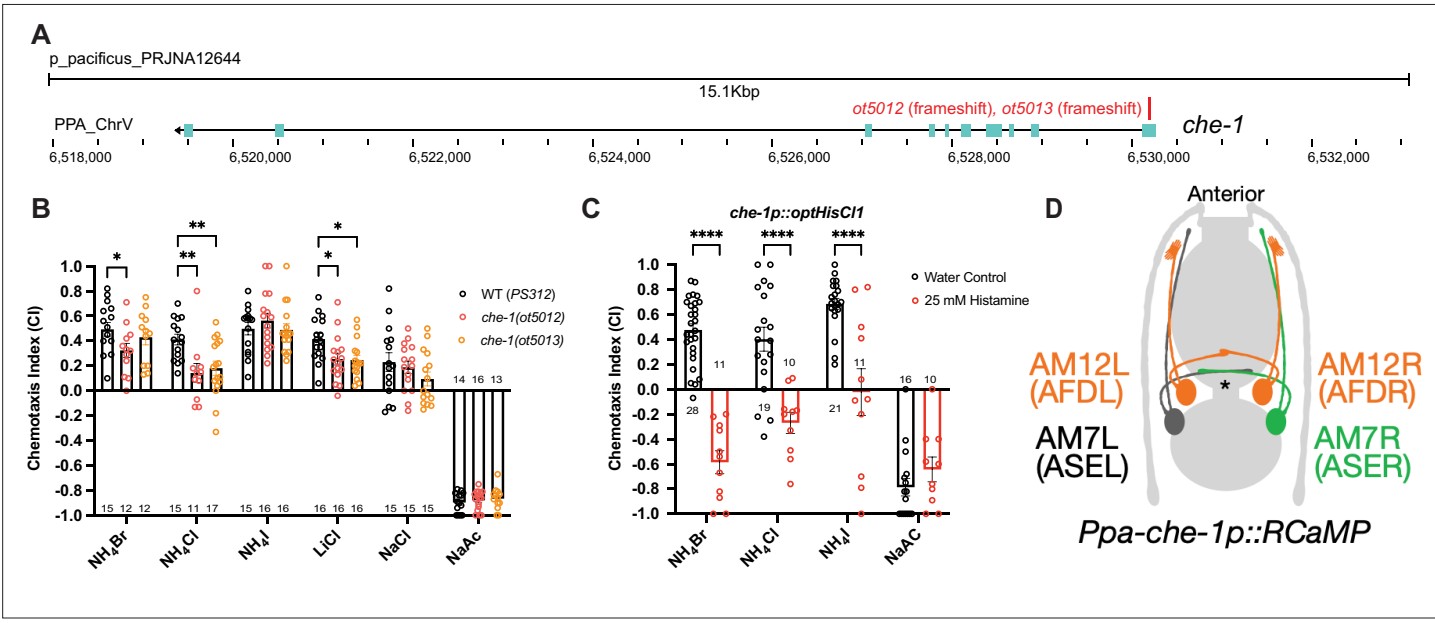

**Figure 3.** *che-1* expressing amphid neurons are required for sensing water-soluble ions in *P. pacificus*. (**A**) The *che-1* locus with CRISPR/Cas9-induced mutations in Exon 1. (**B**) The *che-1* mutants show defects in attraction toward NH$_4$Br, NH$_4$Cl, and LiCl. Sample sizes are indicated below for attractants and above for repellent. (**C**) The *che-1p::HisCl1* animals lose attraction toward NH$_4$Br, NH$_4$Cl, and NH$_4$I in a histamine-dependent manner. Sample sizes are indicated at the base of each bar. (**D**) A schematic map of the *P. pacificus* AM7 (ASE) and AM12 (AFD) amphid neurons that express the *che-1p::RCaMP* used in calcium imaging. The ASE axons are the only amphid axons in *P. pacificus* to cross each other over the dorsal lateral midline contralaterally. **\*\***p < 0.01, **\***p < 0.05, two-way ANOVA with Dunnett's post hoc comparison showing alleles with significant difference to wildtype *P. pacificus* (PS312). **\*\*\*\***p < 0.0001, two-way ANOVA showing significant difference between water control and histamine treatment.

The online version of this article includes the following figure supplement(s) for figure 3:

**Figure supplement 1.** The *che-1p::HisCl1* transgene is necessary for histamine-dependent knockdown of salt attraction.

**Figure supplement 2.** Molecular lesions for *P. pacificus che-1* and *gcy 22.3*.

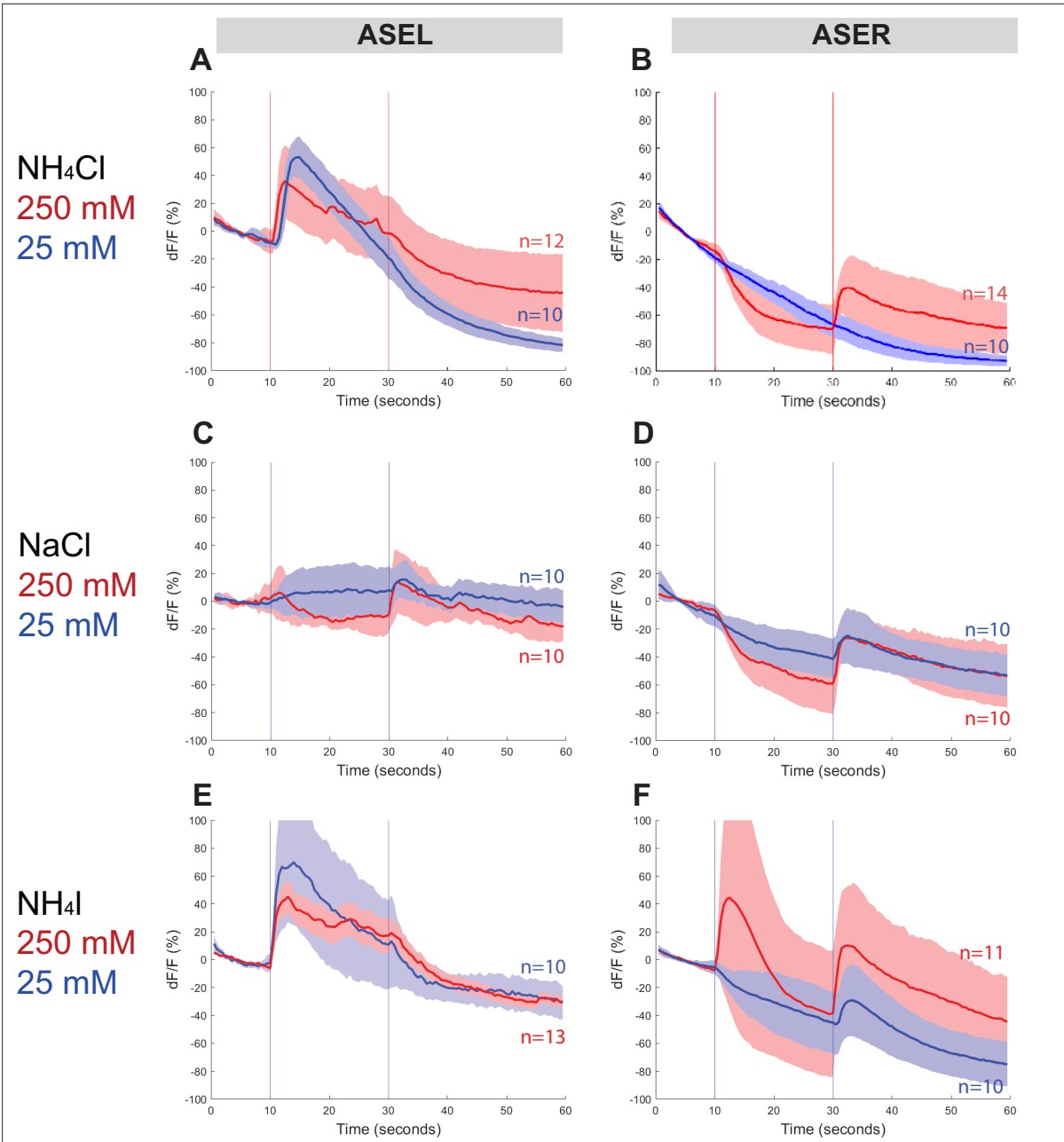

**Figure 4.** ASEL and ASER responses to different concentrations of NH₄Cl, NaCl, and NH₄I. Average percent change in RCaMP fluorescence (d*F/F*) over time (seconds) of tracked ASE (left and right) sensory neurons in *P. pacificus*. Salts were presented at 10 s ('ON', left vertical line) for a duration of 20 s, and then removed ('OFF', right vertical line) for the remaining 30 s; the total recording time was 60 s. (**A, B**) ASEL and ASER neuron responses to high (250 mM, red) compared to low (25 mM, blue) concentrations of NH₄Cl. (**C, D**) ASEL and ASER neuron responses to NaCl. (**E, F**) ASEL and ASER neuron responses to NH₄I. Shaded ribbons represent 95% confidence intervals. Shaded ribbons in (**E, F**) have been cropped to maintain consistent *y*-axes across the plots, allowing for easier comparison. Sample sizes are indicated (*n*).

The online version of this article includes the following figure supplement(s) for figure 4:

**Figure supplement 1.** Negative controls of ASE and AFD neuron responses to green light.

**Figure supplement 2.** Individual traces and heatmaps of ASE responses to various salts.

**Figure supplement 3.** Microfluidic apparatus used for calcium imaging.

## The AFD neurons also respond to salts in *P. pacificus*

The RCaMP line that we used to assess calcium responses in ASE is also expressed in AFD, allowing us to simultaneously examine the calcium responses in the AFD (AM12) neurons (*Figure 5*, *Figure 2—figure supplement 1*, *Figure 5—figure supplements 1 and 2*, *Figure 7—figure supplement 1*). Surprisingly, we detected a distinctly 'ON–OFF' biphasic response to all three salt types at both concentrations. Specifically, although we observed weaker or comparable responses in AFD neurons when compared to either ASE neuron's response toward 250 mM NH$_4$Cl, NaCl, and NH$_4$I (*Figure 5A, B, E, F, I, J*), the AFD responses were more robust than ASER toward 25 mM NH$_4$Cl, NaCl, and NH$_4$I (*Figure 5D, H, L*). Specifically, AFD neurons responded strongly to 25 mM NaCl when neither one of the ASE neurons showed a positive response (*Figure 5G, H*). Averaging the calcium transients separately by AFD left versus right did not result in significant differences in the shape of the neuronal calcium responses, with the exception of the AFDR responses to higher versus lower concentrations of NH$_4$I (*Figure 5—figure supplement 2*, *Figure 7—figure supplement 1*). We have not further pursued whether these AFD responses are a reflection of a direct perception of salt or a secondary consequence of communication of salt-perceptive neurons (like ASE) to AFD.

## A target of *che-1*, the guanylyl cyclase *gcy-22.3*, is required for ASER salt response

We further explored the asymmetric salt perception by the ASE neurons, which in *C. elegans* is largely mediated through distinct receptor-type guanylyl cyclases (rGC proteins, encoded by *gcy* genes), which confer salt specificity via their extracellular domains (*Ortiz et al., 2009*). Our previous genome survey of *Ppa* homologs of *gcy* genes has revealed patterns that made us question whether *Ppa gcy* genes are convincing candidates for lateralized chemotactic responses. Specifically, we noted that ASER-expressed *C. elegans gcy* genes and ASEL-expressed *C. elegans gcy* genes have only expanded in the *Caenorhabditis* genus (*Hong et al., 2019*). One outlier to this pattern is the *Cel-gcy-22* gene, which is expressed in ASER, but has not expanded in *C. elegans* (*Hong et al., 2019*; *Ortiz et al., 2006*). However, this gene has duplicated several times in *P. pacificus*, resulting in 5 putative *Ppa-gcy-22* paralogs (*Hong et al., 2019*). We fused the promoter of one of these paralogs, *Ppa-gcy-22.3*, to *gfp* and found that transgenic animals express GFP exclusively in ASER (*Figure 6A*), identical to the *C. elegans gcy-22* ortholog (*Ortiz et al., 2006*). We confirmed its expression in ASER by analyzing animals that carry both the *Ppa-gcy-22p::gfp* reporter and the *Ppa-che-1p::rfp* reporter, showing a unilateral overlap of these reporters in ASER (*Figure 6B*).

To assess whether *Ppa-gcy-22.3* is a potential effector of *Ppa-che-1* function, we crossed the *gcy-22.3* reporter into *che-1(ot5012)* mutants. We found expression of *gcy-22.3* was eliminated (*Figure 6C*), leading us to conclude that *gcy-22.3* is a potential effector of *che-1* function, identical to its homolog in *C. elegans*.

To determine whether and which of the observed salt responses is mediated by the ASER-expressing *gcy-22.3*, we generated a putative *gcy-22.3 null* mutant through CRISPR/Cas9 genome editing (2 bp complex deletion that introduces a frameshift) (*Figure 7A*) and examined its response to the higher salt concentration. The 'OFF' response to 250 mM NH$_4$Cl was notably abolished in the ASER neuron in the loss-of-function *gcy-22.3* mutant, while the 'ON' response in the ASEL remained intact (*Figure 7B, C*). However, the responses to 250 mM NaCl were not significantly reduced in either the ASEL or ASER neuron in the *gcy-22.3* mutant (*Figure 7D, E*). Furthermore, the *gcy-22.3* mutation also reduced the 'ON' portion of the AFD biphasic response following presentation of 250 mM NH$_4$Cl and NaCl (*Figure 7F, G*). We also examined the behavioral responses toward individual salt ions in *gcy-22.3* mutants, including a second loss-of-function allele *csu182*. Overall, we found no significant differences between wildtype and mutants in responses toward individual ions, and only the attraction to NH$_4$Cl was slightly enhanced in both alleles (*Figure 8*), which was unexpected given the lack of calcium response observed in *gcy-22.3(csu181)* toward NH$_4$Cl. Our findings show that while proper ASE function is critical for salt attraction, defects in individual *gcy* genes do not lead to a major impact on the worms' ability to track toward attractive salts.

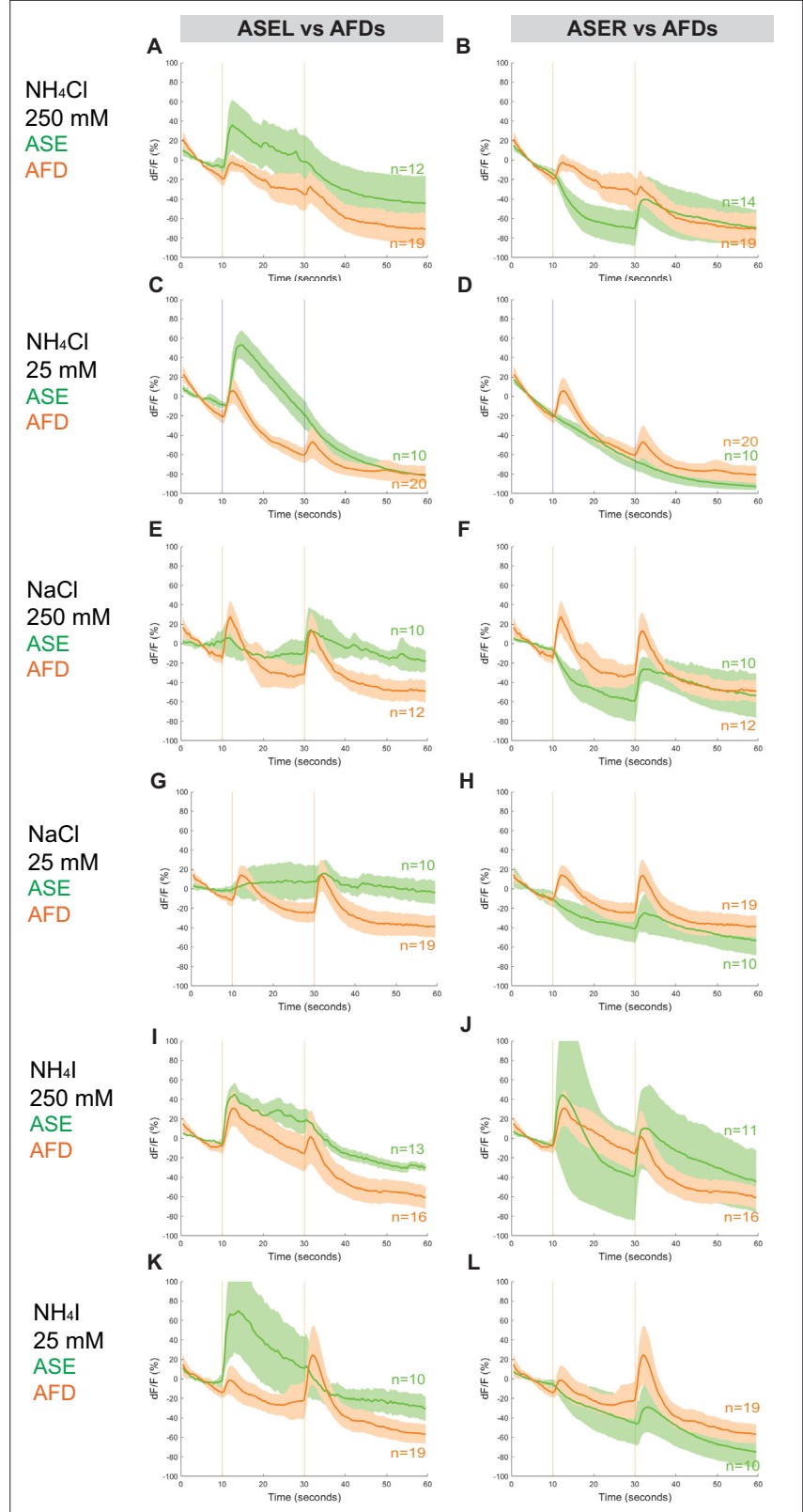

**Figure 5.** Combined AFD responses in comparison to ASE left or right neuron responses to NH$_4$Cl, NaCl, and NH$_4$I. Average percent change in RCaMP fluorescence (d$F$/$F$) over time (seconds) as described in *Figure 4*. Averaged combined AFD (left and right neurons, orange) compared to left or right ASE (green) responses to (**A, B**) 250 mM NH$_4$Cl, (**C, D**) 25 mM NH$_4$Cl, (**E, F**) 250 mM NaCl, (**G, H**) 25 mM NaCl, (**I, J**) 250 mM NH$_4$I, and (**K,**

*Figure 5 continued on next page*

*Figure 5 continued*

**L**) 25 mM NH₄I. Shaded ribbons represent 95% confidence intervals. Shaded ribbons in (**J, K**) have been cropped to maintain consistent *y*-axes across the plots, allowing for easier comparison. Sample sizes are indicated (*n*).

The online version of this article includes the following figure supplement(s) for figure 5:

**Figure supplement 1.** Individual traces and heatmaps of AFD responses to various salts.

**Figure supplement 2.** AFDL/R responses to different concentrations of NH₄Cl, NaCl, and NH₄I.

## Discussion

Our study has revealed several insights into the substrates of evolutionary changes between two distantly related nematode species, *P. pacificus* and *C. elegans*. We used intracellular calcium levels as our readout for neuronal activity with a genetically encoded calcium sensor in two pairs of *che-1*-expressing amphid sensory neurons – the first calcium imaging study in *P. pacificus*. We show that three neuron types (ASE left, ASE right, and AFD neurons) each have distinct calcium responses to specific ion concentrations, revealing diversity at the single neuron level. We have identified the first laterally asymmetric marker in *P. pacificus*, *gcy-22.3p::GFP*, with its expression limited to ASER homolog (AM7). Our unexpected discovery that neuronal asymmetry is present in the ASE homologs between two distantly related nematode species, despite the lack of a *lsy-6* homolog, suggests that functional lateralization in *P. pacificus* may be mediated by a different genetic pathway compared to *C. elegans*.

### *P. pacificus* and *C. elegans* have diverged taste palates

We have shown that while *C. elegans* is attracted to acetate salts, *P. pacificus* avoids these acetates. Previous studies have shown that ammonium acetate (NH₄Ac) is sensed both as a water-soluble compound as well as a volatile odorant and is mediated by different signaling pathways (*Frøkjaer-Jensen et al., 2008*). Like *C. elegans*, it is therefore likely that ammonium and acetate ions involve a different set of neurons in *P. pacificus* (i.e. non-*che-1* expressing neurons), based on the finding that histamine-treated *che-1p::HisCl1* animals did not significantly attenuate their repulsive response to NaAc. We find that sodium chloride is a common attractant for both *P. pacificus* and *C. elegans*, although the magnitude of response is lower than previously published results (*Suzuki et al., 2008*). This result is likely due to differences in generating the salt gradients. Nevertheless, we find that *P. pacificus* neurons have a distinct response to this salt when compared to those observed in *C. elegans* neurons (*Table 1*). The *P. pacificus* ASEL neuron responds to a decrease in NaCl concentration (likely sodium), as evidenced by the 'OFF-2' response profile, whereas the *C. elegans* ASEL neuron responds to an increase in sodium concentration. However, in both nematodes, the ASER neuron responds to a decrease in chloride concentration. Additionally, the *P. pacificus* ASER neuron exhibits a unique ON–OFF response to ammonium iodide, whereas in *C. elegans*, no ON–OFF type of response is seen in the ASE neurons – the *C. elegans* ASER neuron responds only to a decrease in iodide concentration. Based on our calcium imaging and chemotaxis results on the *gcy-22.3* mutants, attraction to ammonium ions (NH₄⁺) is likely mediated in part by the ASER neuron in *P. pacificus*. *Pristionchus* species are entomophilic and most frequently found to be associated with beetles in a necromenic manner and thus insect cadavers could be sources of ammonium in the soil (*Koneru et al., 2016*; *Herrmann et al., 2007*; *Herrmann et al., 2006*; *Fielding et al., 2013*). Additionally, ammonium salts could represent

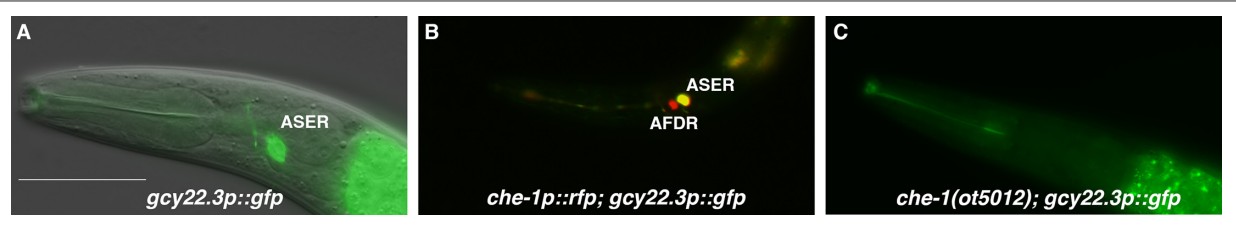

**Figure 6.** The laterally asymmetric expression of *gcy-22.3* is dependent on the zinc finger transcription factor CHE-1. (**A**) The *gcy-22.3::GFP* marker is expressed exclusively in the right ASE neuron (ASER) (*n* > 200). (**B**) The *gcy-22.3::GFP* marker co-localizes with *che-1::RFP* expression in the ASER. (**C**) *gcy-22.3::GFP* expression is absent in the *che-1(ot5012)* mutant (*n* = 55). Anterior is left and dorsal is top. Scale bar in (**A**) represents 50 µm for all panels.

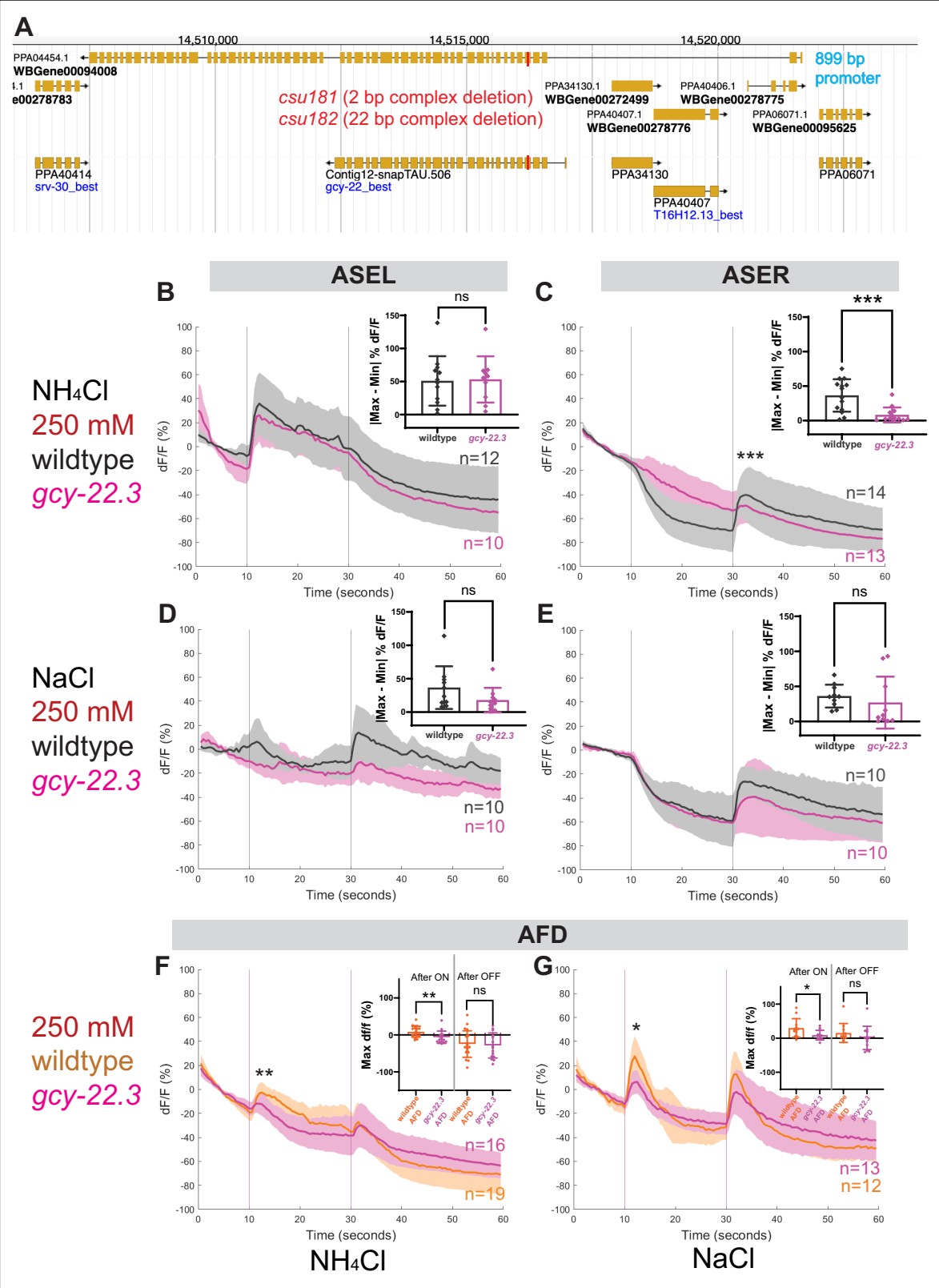

**Figure 7.** ASE and AFD responses in wildtype compared to *gcy-22.3* mutants. Average percent change in RCaMP fluorescence (d*F/F*) over time (seconds) as described in *Figure 4*. (**A**) http://pristionchus.org/ Genome Browser view of the *gcy-22.3* locus with the CRISPR/Cas9-induced mutations indicated with a red bar (Exon 5 of PPA04454 or Exon 4 of Contig12-snapTAU.506). (**B, C**) ASEL and ASER neuron responses to 250 mM NH₄Cl wildtype (gray) and *gcy-22.3* mutants (magenta). (**D, E**) ASEL and ASER neuron responses to 250 mM NaCl in wildtype (gray) and *gcy-22.3* mutants (magenta).

*Figure 7 continued on next page*

*Figure 7 continued*

AFD (combined left and right neurons) responses (**F**) to 250 mM NH$_4$Cl and (**G**) to 250 mM NaCl in wildtype (orange) and *gcy-22.3* mutants (magenta). Shaded ribbons represent 95% confidence intervals. Sample sizes are indicated (*n*). For ASEL/R comparisons, bar plots represent the difference between the minimum % d*F/F* value 10 s pre-stimulus and maximum % d*F/F* value 10 s post-stimulus for either the (**B**) ON or (**C, E**) OFF stimulus. For AFD comparisons, bar plots represent maximum % d*F/F* values 10 s after the ON and OFF stimulus. *p < 0.05, **p < 0.01, ***p < 0.001 indicate significant difference. 'ns' indicate no significant difference. Comparisons between different genotypes were analyzed using unpaired *t*-test or Mann–Whitney test.

The online version of this article includes the following figure supplement(s) for figure 7:

**Figure supplement 1.** AFDL and AFDR responses to NH$_4$Cl and NaCl in wildtype and *gcy-22.3* mutant.

a biological signature of other nematodes that the predatory morph of *P. pacificus* could interpret as prey. In *P. pacificus*, nutritional state has a measurable role in the mouth-form polyphenism decision between predatory and non-predatory morphs (**Piskobulu et al., 2025**). Collectively, our findings highlight the divergence of the *P. pacificus* salt sensory neurons compared to those observed in *C. elegans*.

## *P. pacificus* ASE neurons have narrower sensitivity range

The sensitivity range of the *P. pacificus* ASER responses is significantly less compared to the ASEL responses, as well as to *C. elegans* ASER responses. Whereas the *C. elegans* ASER has a 40-fold sensitivity range in the 'OFF' response to the removal of various concentrations of NaCl (1–40 mM) (**Suzuki et al., 2008**; **Rabinowitch et al., 2014**; **Shindou et al., 2019**; **Watteyne et al., 2020**), the *P. pacificus* ASER showed the 'OFF' response only to 250 mM NH$_4$Cl but not to a tenfold reduction in concentration of NH$_4$Cl (25 mM). For the ASEL in contrast, the response to 25 mM was just as strong as to 250 mM NH$_4$Cl (tenfold) and comparable to the eightfold concentration range observed for *C. elegans* ASEL toward NaCl. Alternatively, the magnitude of these sensitivity differences may be also partially due to differences among calcium indicators (i.e. GCaMP and Cameleon), but multiple

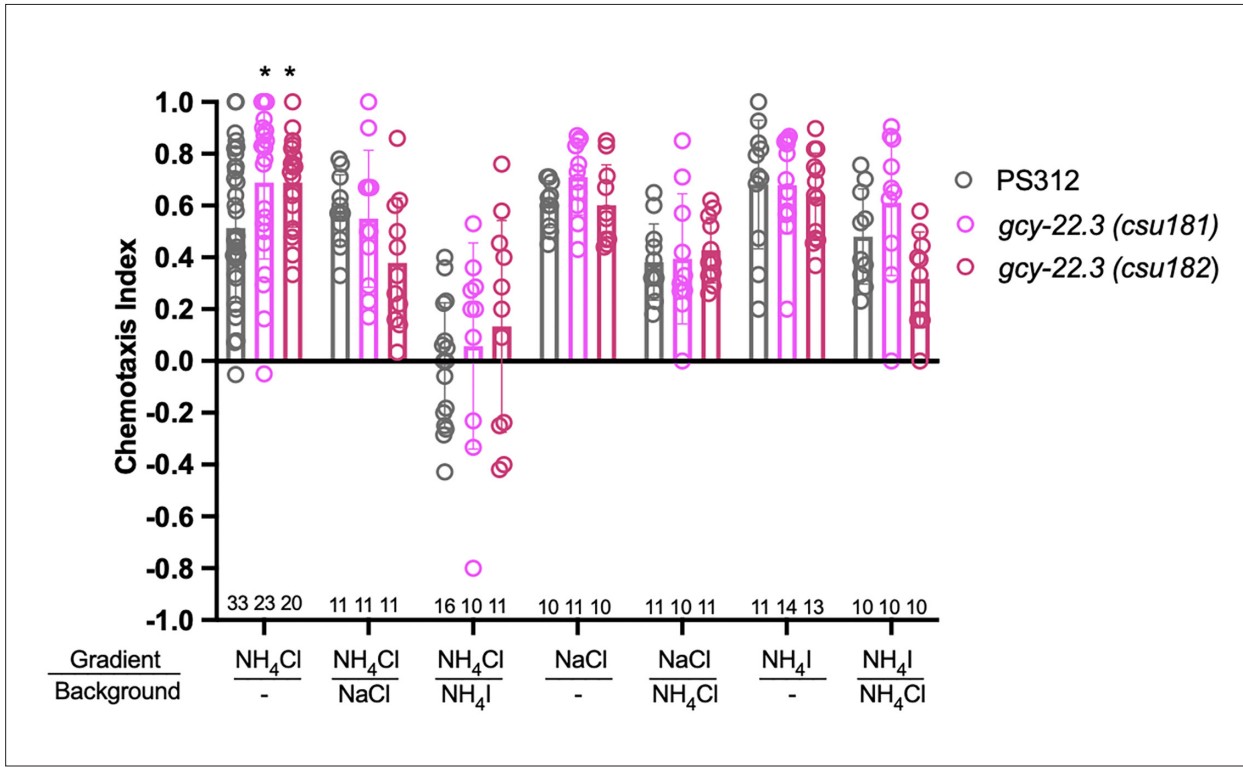

**Figure 8.** Chemotaxis responses to individual ions in *gcy-22.3* mutants. Young adult hermaphrodite responses to salt gradients with or without various salts in background. *p < 0.05. Significant differences were found between wildtype PS312 and *gcy-22.3* mutants to NH$_4$Cl by two-way ANOVA and Dunnett's test. Each assay involves a minimum of 10 animals. Sample sizes for each condition are indicated on the bottom. Error bars denote standard error of the mean.

**Table 1.** Comparison of ASEL/R responses between *P. pacificus* and *C. elegans*.

| | P. pacificus | | C. elegans | |
| --- | --- | --- | --- | --- |
| | **ASEL** | **ASER** | **ASEL** | **ASER** |
| NH₄Cl (1–80 mM) | ON | x | x (*Suzuki et al., 2008*; *Ortiz et al., 2009*) | OFF (*Suzuki et al., 2008*; *Ortiz et al., 2009*) |
| NH₄Cl (81–250 mM) | ON | OFF | ? | ? |
| NaCl (1–80 mM) | x | OFF | ON (*Suzuki et al., 2008*; *Herrmann et al., 2007*; *Herrmann et al., 2006*; *Fielding et al., 2013*) | OFF (*Suzuki et al., 2008*; *Herrmann et al., 2007*; *Herrmann et al., 2006*; *Fielding et al., 2013*) |
| NaCl (81–250 mM) | OFF-2 | OFF | ON (*Herrmann et al., 2006*) | OFF (*Herrmann et al., 2006*) |
| NH₄I (1–80 mM) | ON | OFF | x (*Ortiz et al., 2009*) | OFF (*Ortiz et al., 2009*) |
| NH₄I (81–250 mM) | ON | ON-OFF | ? | ? |

'x' minimal to no response.
'?' represents unknown.

*P. pacificus che-1p::GCaMP* strains did not exhibit sufficient basal fluorescence to allow for image tracking and direct comparison. The narrower sensitivity range in *P. pacificus* chemosensation was also observed for attraction to volatile odors (*Hong and Sommer, 2006*), which span only 10-fold, versus up to 10,000-fold in attractive odors for *C. elegans* (*Bargmann and Horvitz, 1991*).

## *P. pacificus* taste neurons exhibit a unique biphasic response

Although the left-ON and right-OFF responses are conserved in the ASE neurons in both species, the biphasic response of ASER to 250 mM NH₄I has not been observed in either of the ASE neurons toward salts. In *C. elegans*, hyperosmotic stimulus such as 1 M glycerol, or high concentrations or long duration of CuSO₄ exposure both result in biphasic responses by the ASH neurons that sense noxious chemicals (*Wang et al., 2015*; *Chronis et al., 2007*). Olfactory neurons that mediate avoidance behavior such as the AWB neurons can also exhibit a biphasic response to the presence and removal of a high concentration isoamyl alcohol that normally elicits avoidance behavior (*Yoshida et al., 2012*). The neuron-specific response can also be dependent on the concentration of the chemical compound, since the ASER to 25 mM NH₄I was a weak 'OFF' rather than a biphasic one. In contrast to the salt-dependent response types by *P. pacificus* ASE neurons, the *P. pacificus* AFD neurons also show exclusively biphasic responses with various amplitudes. Further characterization will help determine if biphasic responses can also be found in other sensory neuron types in *P. pacificus*, specifically those neurons mediating avoidance behavior.

## AFD are potentially polymodal neurons

Broadly speaking, *C. elegans* chemosensory neurons have been classically characterized as specialized neurons for dedicated modalities such as water-soluble chemicals (ASE), volatile odorants (AWA, AWB, and AWC), noxious chemicals (ASH), pheromones (ADL, ADF, and ASK), and light (ASJ) and temperature (AFD) (*Bargmann, 2006*; *Bargmann et al., 1993*; *Mori and Ohshima, 1995*; *Sengupta et al., 1996*; *Macosko et al., 2009*; *Liu et al., 2010*). Advances in multi-neuron calcium recordings have since shown that a given odor within a certain concentration range is detected by different ensembles of the 12 amphid neuron classes, including the AFD neurons (*Leinwand and Chalasani, 2013*; *Lin et al., 2023*; *Yemini et al., 2021*). Unexpectedly, we found that the *P. pacificus* AFD neurons exhibit a distinctive biphasic response to all three salts tested (NH₄Cl, NaCl, and NH₄I), which differ from the *C. elegans* AFD calcium responses (*Yemini et al., 2021*; *Zaslaver et al., 2015*). Moreover, the loss of the receptor *gcy-22.3* reduced the AFD 'ON' response to NH₄Cl and NaCl, indicating ASER contributes to the AFD response. The strong positive AFD response to 25 mM NaCl that is absent in both ASE neurons further supports the likelihood that AFD receives inputs from other amphid neurons. The integration of thermosensation and chemosensation is important

for memory-regulated behavior. In *C. elegans*, maximum chemotaxis indices toward NH$_4$Cl occurs when there is concordance between cultivation temperature and assay temperature (*Kuhara et al., 2008*). The *C. elegans* AFD neurons are also important for gustatory aversive learning in NaCl avoidance (*Watteyne et al., 2020*). Given the influence of environmental temperature on the *P. pacificus* mouth-form plasticity and the wide range of micro-climates that wild strains of *P. pacificus* have been isolated from *Leaver et al., 2016*; *Sieriebriennikov et al., 2017*; *Leaver et al., 2022*, temperature and taste preferences could be regulated at multiple genetic levels during crucial developmental decisions.

## Changes in the gene regulatory architecture of sensory neuron specification

We found that the key regulator of *Cel* ASE identity, *che-1*, is also expressed in *Ppa* ASE and may play a similar role as a terminal selector in this neuron type, based on its effect on ASE-mediated behavior and regulation of the *Ppa-gcy-22.3* gene. However, the stronger behavioral effect of silencing of *che-1* expressing neurons compared to a *che-1* mutant background could either indicate that *che-1* does not have as broad a role in controlling ASE differentiation in *P. pacificus* versus *C. elegans*. It is possible that a developmental loss of ASE differentiation may result in compensatory changes in the chemosensory system during early development, as was observed in the *C. elegans* mating pheromone response by males (*White et al., 2007*). It is also worthwhile to note that *Cel-gcy-22* stands out for being located on a separate chromosome (Chr. V) compared to the other ASER-type *gcy* genes (*gcy-1*, *gcy-4*, *gcy-5* on Chr. II) as well as the *Cel-gcy-22* mutant having defects in chemoattraction toward a wide range of salt ions (*Ortiz et al., 2009*; *Ortiz et al., 2006*). Given that the five *P. pacificus* *gcy-22*-like paralogs are located on three separate chromosomes (Chr. I, IV, and X), it is likely that they emerged from independent and repeated gene duplication events after the separation of *Caenorhabditis* and *Pristionchus* lineages. Although the promoter fusion reporters of other *gcy* genes have been uninformative due to lack of expression (*Ppa-gcy-22.1*, *Ppa-gcy-7.1*, *Ppa-gcy-7.2*, and *Ppa-gcy-5*), it is likely that ASEL-specific *gcy* genes as well as additional ASER-specific *gcy-22*-like genes exist. Finding other genes with left–right specific expression could help to identify genetic determinants affecting lateral asymmetry.

Unexpectedly, we found that unlike in *C. elegans*, the *Ppa-che-1* gene is also expressed in the AFD neurons. Since we cannot record neural activity in AFD in a *che-1* mutant (the *che-1p::RCaMP* driver fails to be expressed sufficiently in *che-1* mutants due to autoregulation), and do not yet have molecular markers for *Ppa* AFD neurons, we cannot assess whether *che-1* affects AFD neuron differentiation.

Perhaps the most striking difference in the gene regulatory architecture of ASE neuron specification is the apparent lack of the key regulator of ASE asymmetry in *P. pacificus*, the miRNA *lsy-6*. In *C. elegans*, the expression of *lsy-6* exclusively in ASEL is prepatterned via an early embryonic Notch signal (*Cochella and Hobert, 2012*) and serves to downregulate the homeodomain transcription factor *cog-1* in the ASEL neuron (*Johnston et al., 2005*). Through a network of downstream regulatory events, asymmetry of rGCs eventually becomes established (*Hobert, 2014*). *cog-1* and several asymmetrically expressed downstream effectors of *cog-1*, such as the *die-1*, *lim-6*, and *fozi-1* transcription factors are conserved in *P. pacificus,* but whether the function of these factors in controlling *P. pacificus* ASE laterality is conserved remains to be determined. In this context it is intriguing to note that another prominent gene regulatory pathway that is controlled by miRNAs in *C. elegans*, the heterochronic pathway (*let-7s* and *lin-4*), appears to have diverged in *P. pacificus* as well, despite the conservation of the overall physiological readouts of this pathway (temporal patterning of cell lineage divisions) (*Sharma et al., 2024*). It is tempting to speculate that miRNA-meditated regulatory process is particularly labile.

In conclusion, our work illustrates how comparative behavioral and genetic analyses in nematodes is a powerful strategy to uncover substrates of evolutionary change in simple nervous systems.

# Materials and methods

**Key resources table**

| Reagent type (species) or resource | Designation | Source or reference | Identifiers | Additional information |
|---|---|---|---|---|
| Recombinant DNA reagent | pMM5 | This paper | Ppa-che-1pei::optRCaMP | |
| Recombinant DNA reagent | pHC30 | This paper | Ppa-che-1pei::optHisCl1 | |
| Recombinant DNA reagent | pVL2 | This paper | Ppa-gcy-22.3p::GFP | |
| Strain (P. pacificus) | Ppa-che-1p::HisCl1 | This paper | Ppa-che-1pei::optHisCl1; Ppa-egl-20p::turboRFP (promoter with first exon and intron) | csuEx83/RLH336 |
| Strain (P. pacificus) | Ppa-che-1::2xALFA | This paper | Ppa-che-1::ALFA (C-terminal tagged to last exon) | RLH325 |
| Strain (P. pacificus) | Ppa-che-1p::RCaMP; che-1p::GFP | This paper | Ppa-che-1pei::optRCaMP; Ppa-che-1pei::optGFP; Ppa-egl-20p::turboRFP | csuEx93/RLH335 |
| Strain (P. pacificus) | Ppa-ttx-1p::RFP | This paper | Ppa-ttx-1pei::RFP (promoter with first exon and intron) | csuEx96/RLH352 |
| Strain (P. pacificus) | Ppa-ttx-1::2xALFA | This paper | Ppa-ttx-1::ALFA (C-terminal tagged to Exon 18) | RLH280 |

## Nematode strains

*P. pacificus* and other nematode strains were maintained at ~20°C on NGM plates seeded with *E. coli* OP50 for food as described previously (**Cinkornpumin et al., 2014**); these are derived from standard *C. elegans* culture methods (**Brenner, 1974**). *P. pacificus* and other nematode strains used are listed in **Supplementary file 1A**.

## Chemotaxis assays

The assay for assessing response to salt gradients was adapted from *C. elegans* and *P. pacificus* chemotaxis assays (**Hong and Sommer, 2006**; **Bargmann and Horvitz, 1991**; **Ortiz et al., 2006**). Overnight salt gradients were established on 10 cm chemotaxis plates containing 20 ml agar (5 mM KPO$_4$, 1 M CaCl$_2$, 3% Bacto-agar, and 1 mM MgSO$_4$) by adding 10 µl of 2.5 M salt solutions for 16 hr. Alternatively, agar containing 25 mM (NH$_4$Cl and NH$_4$I) or 50 mM (NaCl) were used to test for responses to individual salt ions. Following the establishment of the overnight point gradient, another 4 µl of the same salt solution or water control was added to reinforce the gradient 4 hr before the assay. Just prior to the assay, 1 µl of 1 M sodium azide was added to both the attractive salt (A) and the control (C) spots. *P. pacificus* J4 to adult hermaphrodites from near-saturated cultures were washed 3× with distilled water and collected by centrifuging at 2000 rpm for 2 min. Approximately 200 worms were loaded onto the edge of each assay plate between the gradient sources, and at least 10 combined worms have to reach the scoring arenas to be considered. At least 10 assays constituted each experimental trial, and multiple trials were conducted and averaged for each condition. The chemotaxis index for each end-point assay plate is defined as (A − C)/(A+C). To conduct conditional knockdowns of neurons, 5 M histamine dihydrochloride (Sigma-Aldrich H7250) stock solution in sterilized deionized water (Arrowhead CA) was filter-sterilized and top plated onto the agar plates to a final histamine concentration of 25 mM histamine approximately 10 min before loading the worms to commence the assay. Most assays lasted 3–4 hr at room temperature to allow *P. pacificus* sufficient time to reach the scoring arenas, with ~40% of the animals participating. We excluded an outlier due to likely scoring error (value = –0.64) in the chemotaxis assay for *gcy-22.3(csu182)* on NH$_4$Cl (**Figure 8**). When using the *csuEx83[Ppa-che-1p::optHisCl1]* strain, only animals expressing *Ppa-egl-20p::RFP* tail marker from the extrachromosomal array were scored. Because of *P. pacificus'* strong aversion to acetate, we could not easily assess the individual contributions of salt ions in a saturated background of ammonium acetate as conventionally practiced in *C. elegans* studies (**Ortiz et al., 2009**).

## *Ppa-che-1p::HisCl1* strain

To make the *Ppa-che-1p::optHisCl1*, the *P. pacificus* codon-optimized histamine-gated chloride channel sequence used in *C. elegans* (*HisCl1*) (**Pokala et al., 2014**; **Pokala and Flavell, 2022**) was designed using (https://hallemlab.shinyapps.io/Wild_Worm_Codon_Adapter/; **Bryant and Hallem, 2021**), was custom synthesized (Twist Bioscience), and subsequently inserted behind the *che-1* promoter (3.1 kb containing the first exon and intron) (**Hong et al., 2019**) to create the pHC30 plasmid construct. This

*Ppa-che-1p::optHisCl1* plasmid (2 ng/µl) along with PS312 genomic DNA (80 ng/µl) and *Ppa-egl-20p::RFP* co-injection marker (2 ng/µl) were individually digested with HindIII and assembled as the injection mix to create *csuEx83*.

### *Ppa-che-1p::RCaMP* reporter strain

To make the *Ppa-che-1pei::optRCaMP*, we generated a transgenic worm strain expressing the codon-optimized genetically encoded calcium indicator (GECI), jRCaMP1a, in the neurons of interest (*Kerr et al., 2000*; *Nakai et al., 2001*). jRCaMP1a is an improved red GECI based on mRuby with comparable sensitivity to GCaMP6 (*Dana et al., 2016*). The codon-optimized RCaMP sequence was custom synthesized (Twist Bioscience), and then subcloned using the plasmid pJET1.2/blunt (Thermo Fisher Scientific) to create the pMM2 plasmid construct. Using Gibson Assembly (E2611, New England Biolab), the codon-optimized *RCaMP* sequence was introduced downstream of the *Ppa-che-1* promoter sequence with the first exon and intron sequences to create the pMM5 plasmid construct. Separately, we also generated a *Ppa-che-1pei::optGFP* transcriptional reporter with codon-optimized *GFP* (pMM3) to enable the localization of the *che-1*-expressing neurons during video acquisition (*Han et al., 2020*). The *Ppa-che-1pei::optRCaMP* (2 ng/µl) and *Ppa-che-1pei::optGFP* constructs (1 ng/µl), along with PS312 genomic DNA (80 ng/µl) and *Ppa-egl-20p::RFP* (1.5 ng/µl) were individually digested with HindIII and assembled as the injection mix to create *csuEx93*. Despite multiple attempts, we were unable to generate an equivalent *che-1pei::GCaMP* transgenic line with sufficient basal level of GCaMP expression for a comparison to the RCaMP calcium dynamics.

## Promoter fusion reporter strains

The *Ppa-gcy-22.3p::GFP* construct contains 899 bp of the upstream promoter of the PPA04554 transcript is fused to the codon-optimized GFP (pVL2). A shorter *gcy-22.3* transcript with different first two exons is also predicted, which is more similar in length to other rGC genes in *P. pacificus* (Contig12-snapTAU.506). The *Ppa-ttx-1pei::RFP* construct contains 1935 bp of the upstream promoter along with the first exon and intron sequences of *Ppa-ttx-1* (PPA26714) fused to the codon-optimized RFP, excluding the first two codons (pDC14). pVL2 (2 ng/µl) or pDC14 (2 ng/µl), along with PS312 genomic DNA (80 ng/µl) were individually digested with HindIII to create the injection mixes to generate the independent reporter strains, *csuEx90[Ppa-gcy-22.3p::GFP]*, as well as *Ppa-ttx-1pei::RFP(csuEx94)* and *Ppa-ttx-1pei::RFP(csuEx96)*. *csuEx96* showed less gland cell expression (which can occlude neuronal expression) and stronger AFD expression than *csuEx94*.

## CRISPR mutagenesis generated mutants

CRISPR/Cas9 mutagenesis was used to generate mutations (*Han et al., 2020*; *Nakayama et al., 2020*). crRNA and primer sequences, and induced mutations, are included in *Supplementary file 1C* and .

### *che-1* alleles (PPA01143)

Target crRNA, tracrRNA, and Cas9 nuclease were purchased from IDT Technologies (San Diego, CA). crRNA and tracrRNA were hydrated to 100 µM with IDT Duplex Buffer, and equal volumes of each (0.61 µl) were combined and incubated at 95°C for 5 min, then 25°C for 5 min. Cas9 protein (0.5 µl of 10 µg/µl) was added, then the mix was incubated at 37°C for 10 min. *Ppa-egl-20p::RFP* was used as a co-injection marker. To reach a final total volume of 40 µl, the Cas9–crRNA–tracrRNA complex was combined with pZH009 (*Ppa-egl-20p::RFP*) DNA to reach 50 ng/µl final concentration using nuclease-free water. F$_1$ progeny were screened for the presence of *Ppa-egl-20p::RFP* expression in the tail and candidate F$_1$'s were sequenced to identify heterozygotes (*Nakayama et al., 2020*). *ot5012* has a 4-bp insertion while *ot5013* has an 8-bp complex insertion/deletion, and both mutations cause frameshift mutations and premature stop codons. Each allele was outcrossed two times to wildtype before characterization.

### *gcy-22.3* alleles (PPA04454)

Target crRNA, tracrRNA, and Cas9 nuclease were purchased from IDT Technologies (San Diego, CA). crRNA (RHL1400) and tracrRNA were hydrated to 100 µM with IDT Duplex Buffer, and equal volumes of each (0.61 µl) were combined and incubated at 95°C for 5 min, then 25°C for 5 min. Cas9 protein (0.5 µl of 10 µg/µl) was added, then the mix was incubated at 37°C for 10 min. *Ppa-egl-20p::RFP*

was used as a co-injection marker. To reach a final total volume of 40 µl, the Cas9–crRNA–tracrRNA complex was combined with pZH009 (*Ppa-egl-20p::RFP*) DNA to reach 50 ng/µl final concentration using nuclease-free water. $F_1$ progeny were screened for the presence of *Ppa-egl-20p::RFP* expression in the tail and candidate $F_1$'s were sequenced to identify heterozygotes (*Nakayama et al., 2020*). *csu181* has a 2-bp complex deletion, while *csu182* has a 22-bp complex deletion, both mutations cause frameshifts and premature stop codons. Each allele was outcrossed two times to wildtype before characterization.

## ALFA C-terminal tagging and immunostaining

For induction of site-specific insertions via CRISPR/Cas9-mediated mutagenesis, target crRNA, tracrRNA, and Cas9 nuclease were treated as described above. Single-stranded DNA repair template containing the ALFA nanobody tag (RHL1551 for CHE-1 and RHL1519 for TTX-1) with 35 bp of homology arms on the 5′ and 3′ sides were purchased from IDT. To minimize sequence identity, the two copies of the ALFA sequence contain silent mutations. The crRNA (RHL1396 for *che-1* and RHL1514 for *ttx-1*) and tracrRNA were hydrated to 100 µM with IDT Duplex Buffer, and equal volumes of each (0.61 µl) were combined and incubated at 95°C for 5 min, then 25°C for 5 min. The ALFA tag was C-terminally inserted to the longest *ttx-1* splice form in the last 18th exon (ppa_stranded_DN30925_c0_g1_i5) (Trinity 2016 transcripts can be found on http://pristionchus.org/). $F_1$ animals expressing the co-injection marker *egl-20::optRFP* were lysed and checked by PCR for insertions.

Non-starved healthy cultures of ALFA-tagged CHE-1 (*csu226[Ppa-che-1*::2xALFA]) or ALFA-tagged TTX-1 (*RLH280[Ppa-ttx-1*::2xALFA]) (from six 6 cm plate cultures) were washed with M9 and filtered (Sartorius 84 g/m², Grade 392) and processed as previously described (*Igreja et al., 2022*). In brief, mixed stage worms were fixed overnight at 4°C on a nutator with 500 µl fixation buffer (4% paraformaldehyde in PBS). The worms were then incubated overnight at 37°C with 500 µl 4% β-mercaptoethanol dissolved in 1% Triton X-100 in 0.1 M Tris pH 7.4, and digested in 200 µl of collagenase buffer with 200 units of collagenase type IV (Gibco, Thermo Fisher Scientific, Waltham, MA, USA) at 37°C for ~3.5 hr. The partially digested worms were subsequently washed three times with 500 µl PBST. After collagenase treatment the centrifugation between the washes was done at low speed, 1000 RCF. After washes, the worms were stained in 50 µl 1% BSA in PBST with the primary antibody (1:100 FluoTag-X2 anti-ALFA-AZdye568, N1502, NanoTag Biotechnologies, Göttigen, Germany), overnight at 4°C in a nutator. Following the washes, worms were resuspended in 50 µl VectaShield mounting medium (H-1000, Vector Laboratories, USA) containing DAPI (Molecular Probes, Thermo Fisher Scientific) and gently mounted on freshly prepared 3% Noble agar pads. Images were acquired on a Leica DM6000 microscope.

## Calcium imaging

To conduct calcium imaging, worms were trapped and imaged within a microfluidic PDMS chip while delivering stimuli directly to the nose of an immobilized worm, as previously described (*Chronis et al., 2007*; *Chalasani et al., 2007*). The microfluidic chip was connected to a programmable valve controller (ValveBank) that enables the user to toggle between 'stimulant ON' and 'stimulant OFF' states. Specifically, the ValveBank allows controlled switching of flow from the two outer buffer channels in the chip such that either the control solution (stimulant OFF) or the stimulant solution (stimulant ON) flows over the worm nose (*Figure 4—figure supplement 3*). Buffer solution (for the outer two channels) consisted of M9 buffer with 0.1% Tween-20 and 1 µg/ml fluorescein (*Stiernagle, 2006*; *Liu et al., 2018*). The worm loading solution consisted of M9 buffer with 0.1% Tween-20 and 1.5 mM tetramisole hydrochloride to immobilize the animal. Water-soluble stimulant solutions consisted of the following water-soluble compounds dissolved in nanopure/milli-Q water: ammonium chloride ($NH_4Cl$), ammonium iodide ($NH_4I$), and sodium chloride (NaCl) at concentrations of 250 and 25 mM. 750 mM NaCl elicited very inconsistent responses. The water-soluble stimulant solution was made by diluting the water-soluble compound in nanopure/milli-Q water with 0.1% Tween-20. The control solution consisted of 0.1% Tween-20 nanopure/milli-Q water. Worms were exposed to the control solution, stimulant solution, and then control solution using a 60-s program: 10 s stimulant OFF, 20 s stimulant ON, and 30 s stimulant OFF. As a negative control, animals were exposed only to the control solution, without switching channels, for the duration of the recording. For each animal, the orientation of the nose and vulva were recorded and used as a guide to determine the ventral and dorsal sides of the

worm, and subsequently, the left and right sides of the worm. Accounting for the plane of focus of the neuron pairs as viewed through the microscope, it was then determined whether the imaged neuron was the worm's left or right neuron of each pair. Images were captured using a Zeiss Axio Observer Z1 inverted fluorescence microscope and a pco.panda 4.2 sCMOS camera. Changes in fluorescence intensity were measured in the neurons of interest while the worm was exposed to green light. Images were processed using MetaMorph software version 7.10.5.476. Images were captured at 500 ms exposure time at 2 fps because the baseline fluorescence in *Ppa-che-1p::optRCaMP* worms was too dim to capture viable data using the standard 100 ms exposure time. Baseline $F_0$ was measured as the average background-subtracted fluorescence from the first 9 s of each recording and change in fluorescence intensity was calculated as $dF/F = (F − F_0)/F_0$, as described (*Liu et al., 2018*). Data were analyzed and plotted using custom scripts generated in MATLAB versions R2021a–R2024a. This code is available at https://github.com/honglabcsun/Calcium-Imaging, copy archived at *Mackie and Hong Lab CSUN, 2025*.

For bar plot comparisons between wildtype and *gcy-22.3* mutants, we calculated minimum pre-stimulus and maximum post-stimulus % $dF/F$ values using custom scripts generated in MATLAB. Min–Max data were exported as text files, manually converted and organized into an XLSX file (Microsoft Excel Office16) and imported into Prism GraphPad software (version 10) for data visualization and statistical analysis.

## Nomenclature

Throughout the results section, *P. pacificus* genes will be referred to without the *Ppa-* prefix; if necessary for comparison to another species such as *C. elegans (Cel-)* the *Ppa-* prefix will then be used.

## Materials availability

The plasmids and viable nematode strains will be made available upon request.

## Acknowledgements

This research is funded by NIH SC1GM140970 to RLH, NIH R56MH096881 to SHC. OH is funded by the HHMI. MM and RLH contributed in conception, design, and acquisition of work. VL, HRC, DLC, IMD, NRK, KTQ, and SJC contributed to data acquisition. RLH, OH, and SHC contributed to the analysis and writing of the work. The authors declare that they have no competing interests. We would also like to thank I Martinez and C Igreja for technical assistance, and M Barsegyan for assistance with data acquisition. All data needed to evaluate the conclusions in the paper are present in the paper and the supplementary materials.

## Additional information

### Funding

| Funder | Grant reference number | Author |
|---|---|---|
| National Institutes of Health | SC1GM140970 | Ray L Hong |
| National Institutes of Health | R56MH096881 | Steven J Cook |
| Howard Hughes Medical Institute | | Oliver Hobert |

The funders had no role in study design, data collection, and interpretation, or the decision to submit the work for publication.

### Author contributions

Marisa Mackie, Data curation, Investigation, Methodology, Writing – original draft, Writing – review and editing; Vivian Vy Le, Nicole R Kushnir, Dylan L Castro, Ivan M Dimov, Steven J Cook, Data curation; Heather R Carstensen, Data curation, Methodology; Kathleen T Quach, Resources, Data

curation, Supervision; Oliver Hobert, Supervision, Funding acquisition, Writing – original draft, Writing – review and editing; Sreekanth H Chalasani, Conceptualization, Resources, Supervision, Writing – original draft, Writing – review and editing; Ray L Hong, Conceptualization, Resources, Formal analysis, Supervision, Funding acquisition, Investigation, Visualization, Methodology, Writing – original draft, Project administration, Writing – review and editing

## Author ORCIDs

Marisa Mackie ⓘ https://orcid.org/0009-0001-5549-4453
Heather R Carstensen ⓘ https://orcid.org/0000-0002-2679-3286
Nicole R Kushnir ⓘ https://orcid.org/0009-0008-9040-3669
Steven J Cook ⓘ https://orcid.org/0000-0002-1345-7566
Oliver Hobert ⓘ https://orcid.org/0000-0002-7634-2854
Sreekanth H Chalasani ⓘ https://orcid.org/0000-0003-2522-8338
Ray L Hong ⓘ https://orcid.org/0000-0003-1870-8659

Reviewer #1 (Public review): https://doi.org/10.7554/eLife.103796.3.sa1
Author response https://doi.org/10.7554/eLife.103796.3.sa2

## Additional files

### Supplementary files

MDAR checklist

Supplementary file 1. Nematode strains, plasmids, and primer sequences.

### Data availability

All data generated or analyzed during this study are included in the manuscript and supporting files. Raw data and code is available at https://github.com/honglabcsun/Calcium-Imaging, copy archived at *Mackie and Hong Lab CSUN, 2025*.

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
