## [Editor Report · eLife Assessment]

Mackie and colleagues present a **valuable** comparison of lateralized gustation in two well-studied nematodes. Their results present **convincing** evidence that ASEL/R lateralization exists and is achieved by different means in P. pacificus compared to *C. elegans*. This work will be of interest to neurobiologists interested in how small nervous systems make sense of the environment, and how evolution can take multiple paths to asymmetry within a neuron class.

---

## [Referee Report · Reviewer #1 (Public review)]

Summary:

Mackie and colleagues compare chemosensory preferences between C. elegans and P. pacificus, and the cellular and molecular mechanisms underlying them. The nematodes have overlapping and distinct preferences for different salts. Although P. pacificus lacks the lsy-6 miRNA important for establishing asymmetry of the left/right ASE salt sensing neurons in *C. elegans*, the authors find that P. pacificus ASE homologs achieve molecular (receptor expression) and functional (calcium response) asymmetry by alternative means. This work contributes an important comparison of how these two nematodes sense salts and highlights that evolution can find different ways to establish asymmetry in small nervous systems to optimize the processing of chemosensory cues in the environment.

Strengths:

The authors use clear and established methods to record the response of neurons to chemosensory cues. They were able to show clearly that ASEL/R are functionally asymmetric in P. pacificus, and combined with genetic perturbation establish a role for che-1-dependent gcy-22.3 in the asymmetric response to NH4Cl.

Weaknesses:

The mechanism of lsy-6-independent establishment of ASEL/R asymmetry in P. pacificus remains uncharacterized.

Comments on revisions: Looks good - all the best

---

## [Author Response]

The following is the authors’ response to the original reviews

**Public Reviews:**

**Reviewer #1 (Public review):**
Summary:Mackie and colleagues compare chemosensory preferences between *C. elegans* and P. pacificus, and the cellular and molecular mechanisms underlying them. The nematodes have overlapping and distinct preferences for different salts. Although P. pacificus lacks the lsy-6 miRNA important for establishing asymmetry of the left/right ASE salt-sensing neurons in C. elegans, the authors find that P. pacificus ASE homologs achieve molecular (receptor expression) and functional (calcium response) asymmetry by alternative means. This work contributes an important comparison of how these two nematodes sense salts and highlights that evolution can find different ways to establish asymmetry in small nervous systems to optimize the processing of chemosensory cues in the environment.Strengths:The authors use clear and established methods to record the response of neurons to chemosensory cues. They were able to show clearly that ASEL/R are functionally asymmetric in P. pacificus, and combined with genetic perturbation establish a role for che-1-dependent gcy-22.3 in in the asymmetric response to NH_4_Cl.Weaknesses:The mechanism of lsy-6-independent establishment of ASEL/R asymmetry in P. pacificus remains uncharacterized.

We thank the reviewer for recognizing the novel contributions of our work in revealing the existence of alternative pathways for establishing neuronal lateral asymmetry without the lsy-6 miRNA in a divergent nematode species. We are certainly encouraged now to search for genetic factors that alter the exclusive asymmetric expression of *gcy-22.3*.

**Reviewer #2 (Public review):**
Summary:In this manuscript, Mackie et al. investigate gustatory behavior and the neural basis of gustation in the predatory nematode Pristionchus pacificus. First, they show that the behavioral preferences of P. pacificus for gustatory cues differ from those reported for *C. elegans*. Next, they investigate the molecular mechanisms of salt sensing in P. pacificus. They show that although the C. elegans transcription factor gene che-1 is expressed specifically in the ASE neurons, the P. pacificus che-1 gene is expressed in the Ppa-ASE and Ppa-AFD neurons. Moreover, che-1 plays a less critical role in salt chemotaxis in P. pacificus than C. elegans. Chemogenetic silencing of Ppa-ASE and Ppa-AFD neurons results in more severe chemotaxis defects. The authors then use calcium imaging to show that both Ppa-ASE and Ppa-AFD neurons respond to salt stimuli. Calcium imaging experiments also reveal that the left and right Ppa-ASE neurons respond differently to salts, despite the fact that P. pacificus lacks lsy-6, a microRNA that is important for ASE left/right asymmetry in C. elegans. Finally, the authors show that the receptor guanylate cyclase gene Ppa-gcy-23.3 is expressed in the right Ppa-ASE neuron (Ppa-ASER) but not the left Ppa-ASE neuron (Ppa-ASEL) and is required for some of the gustatory responses of Ppa-ASER, further confirming that the Ppa-ASE neurons are asymmetric and suggesting that Ppa-GCY-23.3 is a gustatory receptor. Overall, this work provides insight into the evolution of gustation across nematode species. It illustrates how sensory neuron response properties and molecular mechanisms of cell fate determination can evolve to mediate species-specific behaviors. However, the paper would be greatly strengthened by a direct comparison of calcium responses to gustatory cues in *C. elegans* and P. pacificus, since the comparison currently relies entirely on published data for C. elegans, where the imaging parameters likely differ. In addition, the conclusions regarding Ppa-AFD neuron function would benefit from additional confirmation of AFD neuron identity. Finally, how prior salt exposure influences gustatory behavior and neural activity in P. pacificus is not discussed.Strengths:(1) This study provides exciting new insights into how gustatory behaviors and mechanisms differ in nematode species with different lifestyles and ecological niches. The results from salt chemotaxis experiments suggest that P. pacificus shows distinct gustatory preferences from *C. elegans*. Calcium imaging from Ppa-ASE neurons suggests that the response properties of the ASE neurons differ between the two species. In addition, an analysis of the expression and function of the transcription factor Ppa-che-1 reveals that mechanisms of ASE cell fate determination differ in *C. elegans* and P. pacificus, although the ASE neurons play a critical role in salt sensing in both species. Thus, the authors identify several differences in gustatory system development and function across nematode species.(2) This is the first calcium imaging study of P. pacificus, and it offers some of the first insights into the evolution of gustatory neuron function across nematode species.(3) This study addresses the mechanisms that lead to left/right asymmetry in nematodes. It reveals that the ASER and ASEL neurons differ in their response properties, but this asymmetry is achieved by molecular mechanisms that are at least partly distinct from those that operate in *C. elegans*. Notably, ASEL/R asymmetry in P. pacificus is achieved despite the lack of a P. pacificus lsy-6 homolog.Weaknesses:(1) The authors observe only weak attraction of *C. elegans* to NaCl. These results raise the question of whether the weak attraction observed is the result of the prior salt environment experienced by the worms. More generally, this study does not address how prior exposure to gustatory cues shapes gustatory responses in P. pacificus. Is salt sensing in P. pacificus subject to the same type of experience-dependent modulation as salt sensing in C. elegans?

We tested if starving animals in the presence of a certain salt will result in those animals avoiding it. However, under our experimental conditions we were unable to detect experiencedependent modulation either in *P. pacificus* or in *C. elegans*.

**Author response image 1. sa2fig1:** 

(2) A key finding of this paper is that the Ppa-CHE-1 transcription factor is expressed in the PpaAFD neurons as well as the Ppa-ASE neurons, despite the fact that Ce-CHE-1 is expressed specifically in Ce-ASE. However, additional verification of Ppa-AFD neuron identity is required. Based on the image shown in the manuscript, it is difficult to unequivocally identify the second pair of CHE-1-positive head neurons as the Ppa-AFD neurons. Ppa-AFD neuron identity could be verified by confocal imaging of the CHE-1-positive neurons, co-expression of Ppa-che1p::GFP with a likely AFD reporter, thermotaxis assays with Ppa-che-1 mutants, and/or calcium imaging from the putative Ppa-AFD neurons.

In the revised manuscript, we provide additional and, we believe, conclusive evidence for our correct identification of Ppa-AFD neuron being another CHE-1 expressing neuron. Specifically, we have constructed and characterized 2 independent reporter strains of *Ppa-ttx-1,* a putative homolog of the AFD terminal selector in *C. elegans*. There are two pairs of *ttx-1p::rfp* expressing amphid neurons. The anterior neuronal pair have finger-like endings that are unique for AFD neurons compared to the dendritic endings of the 11 other amphid neuron pairs (no neuron type has a wing morphology in *P. pacificus*). Their cell bodies are detected in the newly tagged TTX-1::ALFA strain that co-localize with the anterior pair of *che-1::gfp*-expressing amphid neurons (n=15, J2-Adult).

We note that the identity of the posterior pair of amphid neurons differs between the *ttx-1p::rfp* promoter fusion reporter and TTX-1::ALFA strains– the *ttx-1p::rfp* posterior amphid pair overlaps with the *gcy-22.3p::gfp* reporter (ASER) but the TTX-1::ALFA posterior amphid pair do not overlap with the posterior pair of *che-1::gfp*-expressing amphid neurons (n=15). Given that there are 4 splice forms detected by RNAseq (Transcriptome Assembly Trinity, 2016; http://pristionchus.org/), this discrepancy between the *Ppa-ttx-1* promoter fusion reporter and the endogenous expression of the Ppa-TTX-1 C-terminally tagged to the only splice form containing Exon 18 (ppa_stranded_DN30925_c0_g1_i5, the most 3’ exon) may be due to differential expression of different splice variants in AFD, ASE, and another unidentified amphid neuron types.

Although we also made reporter strains of two putative AFD markers, *Ppa-gcy-8.1 (PPA24212)p::gfp; csuEx101* and *Ppa-gcy-8.2 (PPA41407)p::gfp; csuEx100*, neither reporter showed neuronal expression.

(3) Loss of Ppa-che-1 causes a less severe phenotype than loss of Ce-che-1. However, the loss of Ppa-che-1::RFP expression in ASE but not AFD raises the question of whether there might be additional start sites in the Ppa-che-1 gene downstream of the mutation sites. It would be helpful to know whether there are multiple isoforms of Ppa-che-1, and if so, whether the exon with the introduced frameshift is present in all isoforms and results in complete loss of Ppa-CHE-1 protein.

According to http://pristionchus.org/ (Transcriptome Assembly Trinity), there is only a single detectable splice form by RNAseq. Once we have a Ppa-AFD-specific marker, we would be able to determine how much of the AFD terminal effector identify (e.g. expression of *gcy-8* paralogs) is effected by the loss of *Ppa-che-1* function.

(4) The authors show that silencing Ppa-ASE has a dramatic effect on salt chemotaxis behavior. However, these data lack control with histamine-treated wild-type animals, with the result that the phenotype of Ppa-ASE-silenced animals could result from exposure to histamine dihydrochloride. This is an especially important control in the context of salt sensing, where histamine dihydrochloride could alter behavioral responses to other salts.

We have inadvertently left out this important control. Because the HisCl1 transgene is on a randomly segregating transgene array, we have scored worms with and without the transgene expressing the co-injection marker (*Ppa-egl-20p::rfp,* a marker in the tail) to show that the presence of the transgene is necessary for the histamine-dependent knockdown of NH_4_Br attraction. This control is added as Figure S2.

(5) The calcium imaging data in the paper suggest that the Ppa-ASE and Ce-ASE neurons respond differently to salt solutions. However, to make this point, a direct comparison of calcium responses in *C. elegans* and P. pacificus using the same calcium indicator is required. By relying on previously published C. elegans data, it is difficult to know how differences in growth conditions or imaging conditions affect ASE responses. In addition, the paper would be strengthened by additional quantitative analysis of the calcium imaging data. For example, the paper states that 25 mM NH_4_Cl evokes a greater response in ASEL than 250 mM NH_4_Cl, but a quantitative comparison of the maximum responses to the two stimuli is not shown.

We understand that side-by-side comparisons with *C. elegans* using the same calcium indicator would lend more credence to the differences we observed in *P. pacificus* versus published findings in *C. elegans* from the past decades, but are not currently in a position to conduct these experiments in parallel.

(6) It would be helpful to examine, or at least discuss, the other P. pacificus paralogs of Ce-gcy22. Are they expressed in Ppa-ASER? How similar are the different paralogs? Additional discussion of the Ppa-gcy-22 gene expansion in P. pacificus would be especially helpful with respect to understanding the relatively minor phenotype of the Ppa-gcy-22.3 mutants.

In *P. pacificus*, there are 5 *gcy-22*-like paralogs and 3 *gcy-7*-like paralogs, which together form a subclade that is clearly distinct from the 1-1 *Cel-gcy-22, Cel-gcy-5,* and *Cel-gcy-7* orthologs in a phylogenetic tree containing all rGCs in *P. pacificus, C. elegans, and C. briggssae* (Hong et al, eLife, 2019). In Ortiz et al (2006 and 2009), *Cel-gcy-22* stands out from other ASER-type gcy genes (*gcy-1, gcy-4, gcy-5*) in being located on a separate chromosome (Chr. V) as well as in having a wider range of defects in chemoattraction towards salt ions. Given that the 5 *P. pacificus gcy-22*-like paralogs are located on 3 separate chromosomes without clear synteny to their C. elegans counterparts, it is likely that the *gcy-22* paralogs emerged from independent and repeated gene duplication events after the separation of these *Caenorhabditis* and *Pristionchus* lineages. Our reporter strains for two other *P. pacificus gcy-22*-like paralogs either did not exhibit expression in amphid neurons (*Ppa-gcy-22.1p::GFP*) or exhibited expression in multiple neuron types in addition to a putative ASE neuron (*Ppa-gcy-22.4p::GFP*). We have expanded the discussion on the other *P. pacificus gcy-22* paralogs.

(7) The calcium imaging data from Ppa-ASE is quite variable. It would be helpful to discuss this variability. It would also be helpful to clarify how the ASEL and ASER neurons are being conclusively identified during calcium imaging.

For each animal, the orientation of the nose and vulva were recorded and used as a guide to determine the ventral and dorsal sides of the worm, and subsequently, the left and right sides of the worm. Accounting for the plane of focus of the neuron pairs as viewed through the microscope, it was then determined whether the imaged neuron was the worm’s left or right neuron of each pair. We added this explanation to the Methods.

(8) More information about how the animals were treated prior to calcium imaging would be helpful. In particular, were they exposed to salt solutions prior to imaging? In addition, the animals are in an M9 buffer during imaging - does this affect calcium responses in Ppa-ASE and Ppa-AFD? More information about salt exposure, and how this affects neuron responses, would be very helpful.

Prior to calcium imaging, animals were picked from their cultivation plates (using an eyelash pick to minimize bacteria transfer) and placed in loading solution (M9 buffer with 0.1% Tween20 and 1.5 mM tetramisole hydrochloride, as indicated in the Method) to immobilize the animals until they were visibly completely immobilized.

(9) In Figure 6, the authors say that Ppa-gcy-22.3::GFP expression is absent in the Ppa-che1(ot5012) mutant. However, based on the figure, it looks like there is some expression remaining. Is there a residual expression of Ppa-gcy-22.3::GFP in ASE or possibly ectopic expression in AFD? Does Ppa-che-1 regulate rGC expression in AFD? It would be helpful to address the role of Ppa-che-1 in AFD neuron differentiation.

In Figure 6C, the green signal is autofluorescence in the gut, and there is no GFP expression detected in any of the 55 *che-1(-)* animals we examined. We are currently developing AFDspecific rGC markers (*gcy-8* homologs) to be able to examine the role of Ppa-CHE-1 in regulating AFD identity.

**Recommendations for the authors:**

**Reviewer #1 (Recommendations for the authors):**
(1) Abstract: 'how does sensory diversity prevail within this neuronal constraint?' - could be clearer as 'numerical constraint' or 'neuron number constraint'.

We have clarified this passage as ‘…constraint in neuron number’.

(2) 'Sensory neurons in the Pristionchus pacificus' - should get rid of the 'the'.

We have removed the ‘the’.

(3) Figure 2: We have had some good results with the ALFA tag using a similar approach (tagging endogenous loci using CRISPR). I'm not sure if it is a Pristionchus thing, or if it is a result of our different protocols, but our staining appears stronger with less background. We use an adaptation of the Finney-Ruvkin protocol, which includes MeOH in the primary fixation with PFA, and overcomes the cuticle barrier with some LN2 cracking, DTT, then H2O2. No collagenase. If you haven't tested it already it might be worth comparing the next time you have a need for immunostaining.

We appreciate this suggestion. Our staining protocol uses paraformaldehyde fixation. We observed consistent and clear staining in only 4 neurons in CHE-1::ALFA animals but more background signals from TTX-1::ALFA in Figure 2I-J in that could benefit from improved immunostaining protocol.

(4) Page 6: 'By crossing the che-1 reporter transgene into a che-1 mutant background (see below), we also found that che-1 autoregulates its own expression (Figure 2F), as it does in *C. elegans*' - it took me some effort to understand this. It might make it easier for future readers if this is explained more clearly.

We understand this confusion and have changed the wording along with a supporting table with a more detailed account of *che-1p::RFP* expression in both ASE and AFD neurons in wildtype and *che-1(-)* backgrounds in the Results.

(5) Line numbers would make it easier for reviewers to reference the text.

We have added line numbers.

(6) Page 7: is 250mM NH_4_Cl an ecologically relevant concentration? When does off-target/nonspecific activation of odorant receptors become an issue? Some discussion of this could help readers assess the relevance of the salt concentrations used.

This is a great question but one that is difficult to reconcile between experimental conditions that often use 2.5M salt as point-source to establish salt gradients versus ecologically relevant concentrations that are very heterogenous in salinity. Efforts to show *C. elegans* can tolerate similar levels of salinity between 0.20-0.30 M without adverse effects have been recorded previously (Hu *et al*., Analytica Chimica Acta 2015; Mah *et al.* Expedition 2017).

(7) It would be nice for readers to have a short orientation to the ecological relevance of the different salts - e.g. why Pristionchus has a particular taste for ammonium salts.

*Pristionchus* species are entomophilic and most frequently found to be associated with beetles in a necromenic manner. Insect cadavers could thus represent sources of ammonium in the soil. Additionally, ammonium salts could represent a biological signature of other nematodes that the predatory morphs of *P. pacificus* could interpret as prey. We have added the possible ecological relevance of ammonium salts into the Discussion.

(8) Page 11: 'multiple P. pacificus che-1p::GCaMP strains did not exhibit sufficient basal fluorescence to allow for image tracking and direct comparison'. 500ms exposure to get enough signal from RCaMP is slow, but based on the figures it still seems enough to capture things. If image tracking was the issue, then using GCaMP6s with SL2-RFP or similar in conjunction with a beam splitter enables tracking when the GCaMP signal is low. Might be an option for the future.

These are very helpful suggestions and we hope to eventually develop an improved *che1p::GCaMP* strain for future studies.

(9) Sometimes *C. elegans* genes are referred to as '*C. elegans* [gene name]' and sometimes 'Cel [gene name]'. Should be consistent. Same with Pristionchus.

We have now combed through and corrected the inconsistencies in nomenclature.

(10) Pg 12 - '...supports the likelihood that AFD receives inputs, possibly neuropeptidergic, from other amphid neurons' - the neuropeptidergic part could do with some justification.

Because the AFD neurons are not exposed directly to the environment through the amphid channel like the ASE and other amphid neurons, the calcium responses to salts detected in the AFD likely originate from sensory neurons connected to the AFD. However, because there is no synaptic connection from other amphid neurons to the AFD neurons in *P. pacificus* (unlike in *C. elegans*; Hong et al, eLife, 2019), it is likely that neuropeptides connect other sensory neurons to the AFDs. To avoid unnecessary confusion, we have removed “possibly neuropeptidergic.”

(11) Pg16: the link to the Hallam lab codon adaptor has a space in the middle. Also, the paper should be cited along with the web address (Bryant and Hallam, 2021).

We have now added the proper link, plus in-text citation. https://hallemlab.shinyapps.io/Wild_Worm_Codon_Adapter/ (Bryant and Hallem, 2021)

Full citation:

Astra S Bryant, Elissa A Hallem, The Wild Worm Codon Adapter: a web tool for automated codon adaptation of transgenes for expression in non-*Caenorhabditis* nematodes, G3 Genes|Genomes|Genetics, Volume 11, Issue 7, July 2021, jkab146, https://doi.org/10.1093/g3journal/jkab146

**Reviewer #2 (Recommendations for the authors):**
(1) In Figure 1, the legend states that the population tested was "J4/L4 larvae and young adult hermaphrodites," whereas in the main text, the population was described as "adult hermaphrodites." Please clarify which ages were tested.

We have tested J4-Adult stage hermaphrodites and have made the appropriate corrections in the text.

(2) The authors state that "in contrast to *C. elegans*, we find that P. pacificus is only moderately and weakly attracted to NaCl and LiCl, respectively." However, this statement does not reflect the data shown in Figure 1, where there is no significant difference between C. elegans and P. pacificus - both species show at most weak attraction to NaCl.

Although there is no statistically significant difference in NaCl attraction between *P. pacificus* and *C. elegans*, NaCl attraction in *P. pacificus* is significantly lower than its attraction to all 3 ammonium salts when compared to *C. elegans*. We have rephrased this statement as relative differences in the Results and updated the Figure legend.

(3) In Figure 1, the comparisons between *C. elegans* and P. pacificus should be made using a two-way ANOVA rather than multiple t-tests. Also, the sample sizes should be stated (so the reader does not need to count the circles) and the error bars should be defined.

We performed the 2-way ANOVA to detect differences between *C. elegans* and *P. pacificus* for the same salt and between salts within each species. We also indicated the sample size on the figure and defined the error bars.

Significance:

For comparisons of different salt responses within the same species:

- For *C. elegans*, NH_4_Br vs NH_4_Cl (**p<0.01), NH_4_Cl vs NH_4_I (* p<0.05), and NH_4_Cl vs NaCl (* p<0.05). All other comparisons are not significant.

- For *P. pacificus*, all salts showed (****p<0.0001) when compared to NaAc and to NH_4_Ac, except for NH_4_Ac and NaAc compared to each other (ns). Also, NH_4_Cl showed (*p<0.05) and NH_4_I showed (***p<0.001) when compared with LiCl and NaCl. All other comparisons are not significant.

For comparisons of salt responses between different species (N2 vs PS312):

- NH_4_I and LiCl (*p<0.05); NaAc and NH_4_Ac (****p<0.0001)

(4) It might be worth doing a power analysis on the data in Figure 3B. If the data are underpowered, this might explain why there is a difference in NH_4_Br response with one of the null mutants but not the other.

For responses to NH_4_Cl, since both *che-1* mutants (rather than just one) showed significant difference compared to wildtype, we conducted a power analysis based on the effect size of that difference (~1.2; large). Given this effect size, the sample size for future experiments should be 12 (ANOVA).

For responses to NH_4_Br and given the effect size of the difference seen between wildtype (PS312) and ot5012 (~0.8; large), the sample size for future experiments should be 18 (ANOVA) for a power value of 0.8. Therefore, it is possible that the sample size of 12 for the current experiment was too small to detect a possible difference between the ot5013 alleles and wildtype.

(5) It would be helpful to discuss why silencing Ppa-ASE might result in a switch from attractive to repulsive responses to some of the tested gustatory cues.

For similar assays using *Ppa-odr-3p::HisCl1*, increasing histamine concentration led to decreasing C.I. for a given odorant (myristate, a *P. pacificus*-specific attractant). It is likely that the amount of histamine treatment for knockdown to zero (i.e. without a valence change) will differ depending on the attractant.

(6) The statistical tests used in Figure 3 are not stated.

Figure 3 used Two-way ANOVA with Dunnett’s post hoc test. We have now added the test in the figure legend.

(7) It would be helpful to examine the responses of ASER to the full salt panel in the Ppa-gcy-22.3 vs. wild-type backgrounds.

We understand that future experiments examining neuron responses to the full salt panel for wildtype and *gcy-22.3* mutants would provide further information about the salts and specific ions associated with the GCY-22.3 receptor. However, we have tested a broader range of salts (although not yet the full panel) for behavioral assays in wildtype vs *gcy-22.3* mutants, which we have included as part of an added Figure 8.

(8) The controls shown in Figure S1 may not be adequate. Ideally, the same sample size would be used for the control, allowing differences between control worms and experimental worms to be quantified.

Although we had not conducted an equal number of negative controls using green light without salt stimuli due to resource constraints (6 control vs ~10-19 test), we provided individual recordings with stimuli to show that conditions we interpreted as having responses rarely showed responses resembling the negative controls. Similarly, those we interpreted as having no responses to stimuli mostly resembled the no-stimuli controls (*e.g.* WT to 25 mM NH_4_Cl, *gcy22.3* mutant to 250 mM NH_4_Cl).

(9) An osmolarity control would be helpful for the calcium imaging experiments.

We acknowledge that future calcium imaging experiments featuring different salt concentrations could benefit from osmolarity controls.

(10) In Figure S7, more information about the microfluidic chip design is needed.

The chip design features a U-shaped worm trap to facilitate loading the worm head-first, with a tapered opening to ensure the worm fits snugly and will not slide too far forward during recording. The outer two chip channels hold buffer solution and can be switched open (ON) or closed (OFF) by the Valvebank. The inner two chip channels hold experimental solutions. The inner channel closer to the worm trap holds the control solution, and the inner channel farther from the worm trap holds the stimulant solution.

We have added an image of the chip in Figure S7 and further description in the legend.

(11) Throughout the manuscript, the discussion of the salt stimuli focuses on the salts more than the ions. More discussion of which ions are eliciting responses (both behavioral and neuronal responses) would be helpful.

In Figure 7, the *gcy-22.3* defect resulted in a statistically significant reduction in response only towards NH_4_Cl but not towards NaCl, which suggests ASER is the primary neuron detecting NH_4_^+^ ions. To extend the description of the *gcy-22.3* mutant defects to other ions, we have added a Figure 8: chemotaxis on various salt backgrounds. We found only a mild increase in attraction towards NH_4_^+^ by both gcy-22.3 mutant alleles, but wild-type in their responses toward Cl^-^, Na^+^, or I^-^. The switch in the direction of change between the behavioral (enhanced) and calcium imaging result (reduced) suggests the behavioral response to ammonium ions likely involves additional receptors and neurons.

Minor comments:(1) The full species name of "*C. elegans*" should be written out upon first use.

We have added ‘*Caenorhabditis elegans*’ to its first mention.

(2) In the legend of Figure 1, "N2" should not be in italics.

We have made the correction.

(3) The "che-1" gene should be in lowercase, even when it is at the start of the sentence.

We have made the correction.

(4) Throughout the manuscript, "HisCl" should be "HisCl1."

We have made these corrections to ‘HisCl1’.

(5) Figure 3A would benefit from more context, such as the format seen in Figure 7A. It would also help to have more information in the legend (e.g., blue boxes are exons, etc.).(6) "Since NH_4_I sensation is affected by silencing of che-1(+) neurons but is unaffected in che-1 mutants, ASE differentiation may be more greatly impacted by the silencing of ASE than by the loss of che-1": I don't think this is exactly what the authors mean. I would say, "ASE function may be more greatly impacted...".

We have changed ‘differentiation’ to ‘function’ in this passage.

(7) In Figure 7F-G, the AFD neurons are referred to as AFD in the figure title but AM12 in the graph. This is confusing.

Thank you for noticing this oversight. We have corrected “AM12” to “AFD”.

(8) In Figure 7, the legend suggests that comparisons within the same genotype were analyzed. I do not see these comparisons in the figure. In which cases were comparisons within the same genotype made?

Correct, we performed additional tests between ON and OFF states within the same genotypes (WT and mutant) but did not find significant differences. To avoid unnecessary confusion, we have removed this sentence.

(9) The nomenclature used for the transgenic animals is unconventional. For example, normally the calcium imaging line would be listed as csuEx93[Ppa-che-1p::optRCaMP] instead of Ppache-1p::optRCaMP(csuEx93).

We have made these corrections to the nomenclature.

(10) Figure S6 appears to come out of order. Also, it would be nice to have more of a legend for this figure. The format of the figure could also be improved for clarity.

We have corrected Figure S6 (now S8) and added more information to the legend.

(11) Methods section, Chemotaxis assays: "Most assays lasted ~3.5 hours at room temperature in line with the speed of P. pacificus without food..." It's not clear what this means. Does it take the worms 3.5 hours to crawl across the surface of the plate?

Correct, *P. pacificus* requires 3-4 hours to crawl across the surface of the plate, which is the standard time for chemotaxis assays for some odors and all salts. We have added this clarification to the Methods.